# Associations Between Indoor Air Pollution and Urinary Volatile Organic Compound Biomarkers in Korean Adults

**DOI:** 10.3390/toxics13080692

**Published:** 2025-08-20

**Authors:** Byung-Jun Cho, Seon-Rye Kim

**Affiliations:** 1Department of Paramedicine, College of Health Science, Kangwon National University, 346 Hwangjo-gil, Dogye-up, Samcheok-si 25949, Republic of Korea; cho6451@gmail.com; 2Institute of Health Medical Education Convergence Research, Kangwon National University, 346 Jungang-ro, Samcheok-si 25913, Republic of Korea

**Keywords:** indoor air pollution, volatile organic compounds, urinary biomarkers, environmental exposure, health disparity, household air pollution

## Abstract

Volatile organic compounds (VOCs) are common indoor air pollutants known to pose significant health risks, yet little is known about how internal exposure varies across populations and environments. This study investigated the associations between indoor air pollutants and urinary VOC biomarkers in a nationally representative sample. We analyzed data from 1880 adults in the eighth Korea National Health and Nutrition Examination Survey (2020–2021) who completed an indoor air quality (IAQ) survey and provided urine samples, assessing the influence of sociodemographic, behavioral, and environmental factors. Indoor concentrations of PM_2.5_, CO_2_, formaldehyde, total VOCs, benzene, ethylbenzene, toluene, xylene, and styrene were measured, alongside the urinary concentrations of nine VOC biomarkers. Associations between pollutants, sociodemographic variables, and biomarkers were evaluated using univariate and multivariable linear regression with Bonferroni correction. Older age, female, lower socioeconomic status (SES), and smoking were associated with higher urinary VOC biomarker concentrations, with smoking showing the strongest associations. Indoor ethylbenzene, styrene, benzene, and CO_2_ were also associated with multiple metabolites. These findings indicated significant associations between household air pollutants and urinary VOC metabolites, with disparities by age, sex, SES, and smoking status, underscoring the importance of targeted IAQ interventions for vulnerable populations.

## 1. Introduction

Exposure to outdoor air pollution has long been recognized as a major public health concern. In recent years, however, attention has increasingly shifted toward indoor air pollution, which poses a significant and often underappreciated health risk in modern living environments [1,2]. People spend over 80% of their time indoors, and this trend intensified during the COVID-19 pandemic [3]. In the United States, adults spend approximately 87% of their time indoors, where pollutant concentrations can be two to five times higher than outdoors [4]. According to the World Health Organization (WHO), an estimated 2.4 billion people globally are exposed to harmful levels of household air pollution [5,6].

Key indoor pollutants include fine particulate matter (PM_2.5_), carbon dioxide (CO_2_), formaldehyde (HCHO), and volatile organic compounds (VOCs) [7,8]. VOCs, in particular, are emitted from numerous sources such as building materials, consumer products, and cleaning agents. Lifestyle factors can also indirectly contribute to VOC exposure; for instance, frequent consumption of packaged foods may involve heating pre-packaged items that emit VOCs or using packaging materials that release chemicals in enclosed spaces. In contrast, direct sources of VOCs—such as air fresheners—are well-documented contributors to indoor pollutant levels [9]. Volatile organic compounds (VOCs) are ubiquitous in both outdoor and indoor environments, and their health impacts are mediated through multiple exposure pathways, including inhalation, dermal absorption, and ingestion. Environmental factors such as humidity and temperature can influence the penetration of VOCs through human skin [10], while simultaneous exposure to complex VOC mixtures may alter the urinary biomarker profiles compared to single-compound exposures [11]. Due to the overlapping metabolic pathways of many VOCs, a single biomarker can reflect exposure to multiple parent compounds, adding complexity to exposure interpretation [12]. Moreover, although indoor environments can be significant sources of VOC exposure, their contribution relative to total exposure varies depending on outdoor and occupational environments, as demonstrated in comparative exposure studies [13]. Indoor ventilation, often inferred from carbon dioxide (CO_2_) levels, may also influence pollutant concentrations, and prior work has linked low-to-moderate CO_2_ concentrations to cognitive performance changes [14]. Moreover, benzene, one of the most representative volatile organic compounds, is classified as a carcinogen [15]. These considerations underscore the necessity of assessing VOC exposures within the broader context of total environmental exposure and ventilation status.

Indoor pollutants not only trigger acute symptoms such as allergic reactions but also contribute to chronic illnesses like cardiovascular disease, chronic obstructive pulmonary disease (COPD), and lung cancer [2,16,17]. Recognizing these risks, the WHO and many national agencies have established standards for indoor air quality (IAQ) management [18]. IAQ is shaped by a combination of internal sources and external factors, including outdoor air quality, ventilation strategies, socioeconomic conditions, and building infrastructure [19,20,21,22,23]. Understanding these complex interactions is vital for effective IAQ control.

Recent studies have confirmed associations between poor IAQ and adverse health outcomes. According to the WHO, over 3 million people die prematurely each year from diseases caused by household air pollution, including ischemic heart disease, lower respiratory infections, stroke, COPD, and lung cancer [23]. Systematic reviews and meta-analyses have also confirmed that VOC exposure increases the risk of respiratory diseases [24,25]. Consequently, internal exposure assessment using biomarkers has become an essential approach in environmental health research.

In Korea, research exploring the relationship between IAQ and human biomarkers remains limited. The eighth Korea National Health and Nutrition Examination Survey (KNHANES) included new variables on IAQ and urinary biomarkers, providing a unique opportunity for large-scale, population-based analysis [26]. This study aims to evaluate the influence of individual characteristics and IAQ factors on urinary VOC biomarker levels. The findings are expected to provide essential evidence to guide policy development and public health interventions aimed at reducing indoor air pollution exposure.

## 2. Materials and Methods

### 2.1. Study Design and Population

This study is a cross-sectional study. This study used data from the Korea National Health and Nutrition Examination Survey (KNHANES), a nationally representative cross-sectional survey conducted by the Korea Disease Control and Prevention Agency (KDCA). Detailed descriptions of the sampling design, participant recruitment, and survey procedures are provided in the KNHANES manuals [26]. We analyzed data from the second and third years of the eighth KNHANES cycle (July 2020–August 2021) to investigate household indoor air quality (IAQ) and biomarkers of exposure to hazardous environmental substances among Korean adults. Households that agreed to the IAQ survey received visits from trained field staff, who conducted environmental measurements and administered structured questionnaires; approximately 1200 households were selected. Adults aged ≥ 19 years who consented to biological sample collection were eligible for biomonitoring. A total of 1968 participants provided urine samples for analysis of volatile organic compound (VOC) exposure biomarkers. After excluding individuals with insufficient information, the final analytic sample comprised 1880 participants (Figure 1). This study is a secondary analysis of publicly available KNHANES data; the authors were not involved in survey design, participant recruitment, fieldwork, or laboratory measurements. (Appendix A
Table A1).

### 2.2. VOC and Indoor Air Pollutant Measurements

Indoor concentrations of VOCs (benzene, toluene, ethylbenzene, xylene, styrene, formaldehyde), fine particulate matter (PM_2.5_), and carbon dioxide (CO_2_) were measured in the participants’ homes using standardized protocols [18,27]. Structured questionnaires captured household characteristics and potential environmental determinants of IAQ. Detailed preparation was mentioned in Appendix A
Table A2.

### 2.3. Biomarker Analysis

Spot urine samples were collected to evaluate internal exposure to environmental pollutants. Nine urinary biomarkers of VOC exposure were analyzed using a validated method: Simultaneous Quantification of VOC Metabolites in Urine by LC-MS/MS (Sciex API 5500, Framingham, MA, USA) operated in multiple reaction monitoring mode. Urinary creatinine concentrations were used to adjust for sample dilution. Following WHO guidelines [28], samples with creatinine concentrations outside the acceptable range (0.3–3.0 g/L) were flagged using a categorical variable (VOC_Ucrea_etc) and excluded from the primary analysis. For biomarker concentrations below the limit of detection (LOD), values were imputed as LOD divided by the square root of two (LOD/√2) [27,28]. Additional information data are provided in Appendix A
Table A3. Method validation was completed prior to study sample analysis: recovery in pooled urine fortified at low, medium, and high levels ranged from 92.0% to 106.4%; intra-assay coefficients of variation (CVs) were 2.1–5.8%; and interassay CVs were 3.2–7.4% across all metabolites, demonstrating high accuracy and precision consistent with international biomonitoring standards. Full validation data are provided in Appendix A
Table A4. Detailed information on biomarker abbreviations and their corresponding LODs are described in Table 1.

### 2.4. Laboratory Procedures and Quality Control

All laboratory preparation and analytical procedures for biomarker measurements were conducted by KDCA following standardized protocols adapted from the U.S. NHANES VOC protocol [29] and aligned with the WHO Air Quality Guidelines [28].

The urinary VOC biomarkers selected for in this study were chosen to enable comparability with previous population-based biomonitoring studies, particularly those utilizing KNHANES and other large-scale datasets. Certain biomarkers were also included for their emerging relevance to public health in Korea. While the combination of biomarkers analyzed here has not previously been investigated collectively within KNHANES, selection criteria emphasized consistency with established protocols and compatibility with prior research. To minimize temporal discrepancies between environmental and biological measurements, IAQ assessments were scheduled within four weeks of urine collection. KDCA implemented a comprehensive quality control (QC) program that included: (1) Verification of instrument performance using low-, medium-, and high-level QC samples at the beginning and end of each analytical batch. (2) Repeat testing of 10% of randomly selected samples to assess reproducibility. (3) Use of certified reference materials to verify accuracy and detect systematic bias. (4) Acceptance criteria of ± 15% for intra-batch precision and ± 20% for inter-batch reproducibility. (5) Standard curve acceptance only if R^2^ ≥ 0.995 for all metabolites [27].

Urinary VOC metabolites were quantified using LC–MS/MS with isotope-labeled internal standards, following KNHANES laboratory guidelines. Accuracy was evaluated by comparing measured concentrations to reference values, calculating percent accuracy, and ensuring coefficients of variation (CV) met acceptance limits. Internal QC included daily verification of instrument performance, calibration, and accuracy checks, while external QC was conducted annually through participation in the German External Quality Assessment Scheme (G-EQUAS) to confirm validity. Re-analysis was performed when acceptance criteria were not met, such as abnormal peak patterns, deviations in internal standard responses, or sample preparation errors. These procedures ensured high precision, accuracy, and reproducibility of VOC metabolite measurements [27].

### 2.5. Covariates

Socioeconomic characteristics were assessed using education (≤elementary school, middle school, high school, ≥college) and household income (low, low-middle, middle-high, high). Additional covariates included age, sex, smoking status (current smoker or non-smoker), alcohol consumption (drinking, non-drinking), time spent at home on weekdays (hours), recent home renovation (within 6 months, yes or no), and use of air purifier (yes or no).

### 2.6. Statistical Analysis

Associations between indoor pollutant concentrations and urinary VOC metabolites were evaluated using univariate and multivariable linear regression models. Prior to regression analyses, boxplots of VOC metabolites were generated across categorical covariates. Because VOC metabolite values showed substantial variation, log-transformed values were used for visualization. In addition, correlation analyses and scatter plots were conducted to explore the relationships between indoor air pollutants and VOC biomarkers. Final multivariable models included the full set of a priori covariates, regardless of statistical significance, unless excluded for high collinearity (variance inflation factor > 5). To address potential inflation of type I error due to multiple hypothesis testing, we applied a Bonferroni correction to all primary association analyses. All reported *p*-values are presented alongside Bonferroni-adjusted significance thresholds, and results that remained significant after adjustment are indicated in the tables. Analyses were performed using SPSS version 25 (IBM Corp., Armonk, NY, USA) and R 4.3.0 (R Foundation for Statistical Computing, Vienna, Austria), with a two-sided *p*-value < 0.05 considered statistically significant.

### 2.7. Ethical Considerations

This study was conducted in accordance with the Declaration of Helsinki. Ethical approval was obtained from the Institutional Review Board of the Korea Disease Control and Prevention Agency (IRB No. KDCA-IRB-2018-01-03-2C-A, 2018-01-03-5C-A). Written informed consent was obtained from all participants prior to data collection, covering both environmental measurements and biological sampling. All participation was voluntary, and data were anonymized to protect confidentiality. The dataset is publicly available through the KNHANES data repository and accessible to qualified researchers for academic purposes.

## 3. Results

### 3.1. Participant Characteristics

Among the 1880 participants, females were slightly older than males and had lower average educational attainment (*p* < 0.001). Males reported higher rates of alcohol consumption and smoking (*p* < 0.001 for both), whereas females spent more time at home on weekdays (*p* < 0.001). Household income distribution and recent housing repairs did not differ significantly by sex, and air purifier use was similar between groups (Table 2). Mean concentrations of indoor air pollutants and urinary VOC biomarkers are presented in Table 3.

### 3.2. Association Between Subjects’ Characteristics and Biomarker Levels

#### 3.2.1. Sex and Biomarker Levels

For most VOC metabolites—except 2MHA and 3,4MHA—mean concentrations were consistently higher in females (Table 4) (Figure 2).

#### 3.2.2. Education Levels and Biomarker Levels

Participants with lower educational attainment exhibited significantly higher levels of all urinary VOC biomarkers (*p* < 0.01), suggesting that socioeconomic status may play a role in exposure to environmental pollutants (Table 5) (Figure 3).

#### 3.2.3. Household Income Levels and Biomarker Levels

Significantly higher concentrations of 3HPMA, SPMA, and DHBMA were observed among participants with low household income levels (*p* < 0.05) (Table 6) (Figure 4).

#### 3.2.4. Usage of Air Purifier and Biomarker Levels

Individuals who did not use air purifiers exhibited significantly elevated levels of nearly all biomarkers (*p* < 0.05), with the exception of 2MHA and 3,4MHA, suggesting that intervention to improve indoor air quality may help reduce internal exposure to environmental pollutants (Table 7) (Figure 5).

#### 3.2.5. Home Repairs and Biomarker Levels

Participants who had completed home renovations within the previous 6 months demonstrated significantly higher concentrations of 3HPMA and DHBMA (*p* < 0.05), suggesting potential exposure to VOCs from building materials or modifications in the indoor environment (Table 8) (Figure 6).

#### 3.2.6. Drinking Status and Biomarker Levels

Individuals who consumed alcohol exhibited lower concentrations of SPMA, BPMA, DHBMA, and BMA, but higher concentrations of 3,4-MHA, suggesting that alcohol consumption may differentially affect metabolic processes or exposure pathways (Table 9) (Figure 7).

#### 3.2.7. Smoking Status and Biomarker Levels

Smokers exhibited significantly higher levels of 3HPMA, PGA, DHBMA, and 3,4-MHA compared to non-smokers (*p* < 0.001), further confirming that smoking serves as a major endogenous source of exposure to key environmental toxicants (Table 10) (Figure 8).

#### 3.2.8. Age and Biomarker Levels

Age exhibited significantly increasing levels of SPMA, PGA, DHBMA, BMA and 3,4-MHA (*p* < 0.01), further confirming that age serves as a major endogenous source of exposure to key environmental toxicants (Table 11).

#### 3.2.9. Time at Home on Weekdays and Biomarker Levels

Time at home on weekdays (hours per day) exhibited significantly increasing levels of SPMA, DHBMA, and 3,4-MHA (*p* < 0.05), further confirming that time at home on weekdays serves as an endogenous source of exposure to environmental toxicants (Table 12).

### 3.3. Multivariable Regression Analysis for VOC Biomarkers

Multiple linear regression analyses identified key determinants of urinary VOC biomarker levels: 3-HPMA was associated with age, smoking status, air purifier use, ethylbenzene, and styrene. PGA was influenced by age, sex, smoking status, BMI, air purifier use, CO_2_, benzene, and ethylbenzene. MA was associated solely with age. 2MHA was associated solely with benzene. SPMA was affected by age, sex, education, air purifier use, carbon dioxide, and xylene. BPMA was influenced by age, smoking status, CO_2_, and ethylbenzene. DHBMA was associated with age, sex, education, income, smoking status, air purifier use, CO_2_, and xylene. BMA was influenced by age, alcohol consumption, and smoking status, while 3,4-MHA was associated with smoking status, HCHO, benzene, and ethylbenzene. Collectively, age, smoking status, air purifier use, and benzene-related compounds (e.g., ethylbenzene, styrene) emerged as common predictors across multiple biomarkers, with SPMA, BPMA, and DHBMA demonstrating particular sensitivity to environmental exposures (Table 13).

### 3.4. Correlation Between Indoor Air Pollutants and Urinary Metabolites

Pearson correlation analyses indicated moderate positive correlations between several indoor air pollutants and their corresponding urinary metabolites, with the strongest association observed for benzene and PGA (r = 0.302, *p* < 0.01) (Table 14). These results suggest that indoor air in residential envrionments may contribute substantially to internal exposure for certain compounds. However, the magnitude of the associations indicates that other exposure pathways (e.g., occupational, commuting, dietary, or dermal routes) also likely play an important role.

### 3.5. Correlation Among Indoor Air Pollutants

Indoor air pollutants, especially VOCs, exhibited positive correlations with each other (r = 0.064–0.748). In contrast, PM_2.5_ showed no significant correlations with any VOCs (Table 15).

## 4. Discussion

This study offers one of the first nationwide analyses examining the association between indoor air pollutants and internal exposure to VOCs in the Korean population, using urinary VOC biomarkers. By linking household air quality measurements with individual-level data, we observed that both environmental and sociodemographic factors were associated with variations in internal exposure levels. Given the cross-sectional nature of this study, these findings should be interpreted as associations rather than evidence of causality.

Urinary VOC metabolites reflect recent, short-term exposure due to their rapid excretion and short biological half-lives. Observed differences by age, gender, and other demographic characteristics may therefore be more plausibly explained by variations in lifestyle, environmental behaviors, and metabolic capacity than by long-term accumulation. While not direct measures of health effects, these biomarkers offer valuable context for identifying potential exposure disparities.

Age and sex differences were evident. Older participants had higher urinary biomarker levels, potentially reflecting differences in recent exposure patterns. This pattern aligns with prior domestic and international studies showing that older populations may be more sensitive to air pollution [30,31,32]. Females consistently showed higher concentrations for several VOC biomarkers, possibly related to indoor activity patterns, use of VOC-containing household or personal care products, or physiological factors such as lower creatinine excretion [31,32], consistent with WHO reports identifying females and children as disproportionately affected by household air pollution [6]. Given that creatinine excretion is generally lower in females due to lower average muscle mass, creatinine-standardized concentrations can appear higher even if absolute exposure is similar. These factors, along with the lack of a strong association between time at home and biomarker levels, suggest that multiple behavioral, environmental, and physiological factors may contribute to the observed sex differences [6,11,31,32].

Socioeconomic disparities were also associated with biomarker concentrations. Participants with lower education and income levels had higher urinary VOC concentrations, potentially due to older housing, inadequate ventilation, and limited access to air purification [19,22,33]. In our analysis, benzene, toluene, and ethylbenzene concentrations were higher in lower-SES households, supporting a pathway through residential environments, though occupational and community exposures may also play a role.

Smoking emerged as a major determinant of internal VOC burden. Smokers had substantially elevated urinary VOC metabolites compared to non-smokers, consistent with prior evidence identifying tobacco smoke as a potent VOC source [34,35]. Combined exposure to cigarette smoke and VOCs significantly increases biomarkers of respiratory inflammation and oxidative stress [34]. Prior studies have further linked indoor air pollution to a range of adverse health outcomes [2,5,24,25,31], and our findings underscore the compounded impact of smoking and environmental VOC exposure.

Several indoor pollutants, including ethylbenzene, styrene, benzene, and CO_2_, were associated with urinary VOC biomarker concentrations. These compounds, emitted from household products and building materials, may undergo metabolic activation leading to systemic inflammation or carcinogenesis [7,8,9,11,15,20,22]. The association between elevated VOC biomarker concentrations and recent home renovations highlights risks from synthetic materials and adhesives. Benzene, classified as a Group 1 carcinogen by the International Agency for Research on Cancer (IARC), further reinforces the importance of minimizing indoor VOC exposure [15].

Interestingly, higher indoor CO_2_ concentrations were sometimes associated with lower biomarker levels, likely reflecting differences in indoor air ventilation or occupancy patterns rather than direct physiological mechanisms [11,14,33,36]. In the SPMA model, although CO_2_ was statistically significant, the effect size was minimal, highlighting the need to consider both statistical and public health relevance.

Interpreting urinary VOC metabolite concentrations requires caution, as many metabolites are non-specific and may originate from multiple compounds. This complexity is amplified in low-level, mixed-source exposure settings. Dermal absorption and co-exposures can also influence biomarker profiles [10,11,12,13,14]. Indoor CO_2_ concentrations, often used as an indirect indicator of ventilation, have also been linked to cognitive performance, highlighting the need to consider indoor ventilation status when interpreting these associations [11,12,14].

Internationally, our findings mirror NHANES, GerES, and HBM4EU results in showing strong associations between smoking, SES, and VOC metabolites. However, certain VOC concentrations here were lower than those reported in NHANES but comparable to GerES, potentially reflecting differences in housing, indoor ventilation, building materials, and regulations [27,37,38].

In this study, several indoor VOC concentrations were associated with corresponding urinary metabolites. However, total VOC exposure derives from multiple pathways—including occupational, commuting, and outdoor sources—making it impossible to precisely quantify the contribution of indoor residential air. However, these findings reinforce WHO guidance advocating targeted protections for vulnerable groups [6,32,39]. Elevated levels of 3HPMA and BPMA in our sample may reflect unique indoor environmental exposures or lifestyle and housing differences [8,15,32,34].

To our knowledge, this is the first study to integrate nationally representative biomonitoring data from the KNHANES with direct household air quality measurements of VOCs, PM_2.5_, CO_2_, and formaldehyde. This combined approach enables real-world, population-based evaluation of internal exposure, accounting for both individual and residential environmental determinants.

Several limitations should be noted. First, the cross-sectional design restricts causal inference. Despite efforts to temporally align air quality measurements with urine sampling, fluctuations in daily exposure may not be fully captured. The IAQ assessment was based on a single household measurement, which may not adequately represent individuals’ long-term exposure in their daily living environments. Second, while we adjusted for several potential confounders, residual confounding from unmeasured factors such as occupational VOC exposure, commuting patterns, dietary VOC intake, and other environmental sources remains possible. The absence of detailed occupational exposure data, in particular, limits our ability to distinguish between residential and workplace contributions to internal VOC burden. Third, the specificity of each urinary biomarker must be interpreted with caution, particularly in environmental exposure settings. Xenobiotics often undergo multiple and overlapping metabolic pathways, meaning that a single parent compound may yield multiple metabolites, and a given metabolite may originate from several different parent compounds. In occupational settings with high single-substance exposures, certain metabolites may serve as relatively specific markers. However, in general population settings, where multiple low-level exposures occur simultaneously, such specificity is reduced. For each biomarker in this study, we have summarized known metabolic pathways and, where available, approximate proportions of parent compounds metabolized to the measured metabolite. Fourth, the KNHANES questionnaire did not collect information on the specific filtration mechanisms employed (e.g., high-efficiency particulate air [HEPA] filters vs. activated carbon or other adsorbents). As a result, it is not possible to determine whether the reported devices were primarily effective for particulate matter, gaseous pollutants, or both. This lack of specificity limits our ability to draw conclusions regarding the potential impact of air purifiers on indoor PM_2.5_ and VOC concentrations. Future studies should adopt longitudinal designs, integrate real-time IAQ assessments with biomonitoring, and inform policy, linking IAQ improvements to housing, urban planning, and chronic disease prevention strategies.

In conclusion, this study highlights notable associations between indoor pollutants and urinary VOC metabolites in the Korean population, with disparities by age, sex, SES, and smoking status. These findings support targeted IAQ interventions for vulnerable subgroups and underscore the value of linking environmental monitoring with population biomonitoring to inform public health strategies.

## 5. Conclusions

In this nationally representative analysis, household indoor air pollutants—including benzene-related compounds—were associated with urinary VOC metabolites, reflecting recent internal exposures. Age, sex, SES, and smoking status emerged as key determinants, highlighting vulnerable subgroups for targeted intervention. The integration of direct IAQ measurements with biomonitoring provides a robust framework for assessing environmental exposures in real-world settings.

While the cross-sectional design precludes causal inference, the findings point to multiple exposure pathways beyond the home environment and emphasize the need for comprehensive indoor air quality management, particularly in lower-SES and high-smoking populations. Future longitudinal studies should incorporate repeated IAQ and biomonitoring data, along with detailed occupational and behavioral assessments, to better characterize sources and refine prevention strategies.

## Figures and Tables

**Figure 1 toxics-13-00692-f001:**
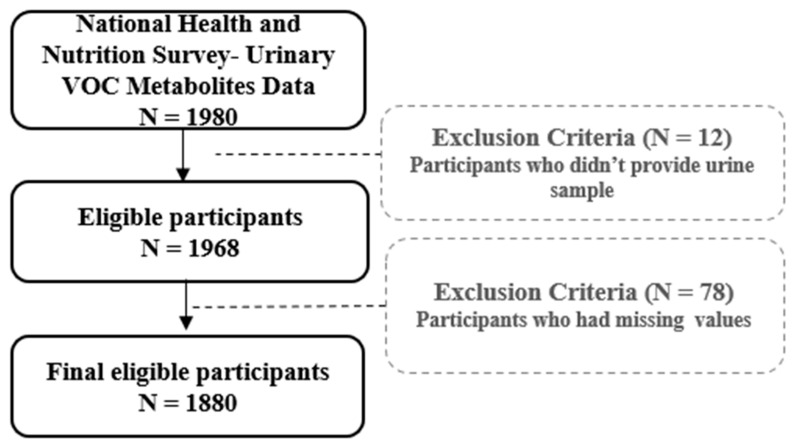
Flowchart of subjects.

**Figure 2 toxics-13-00692-f002:**
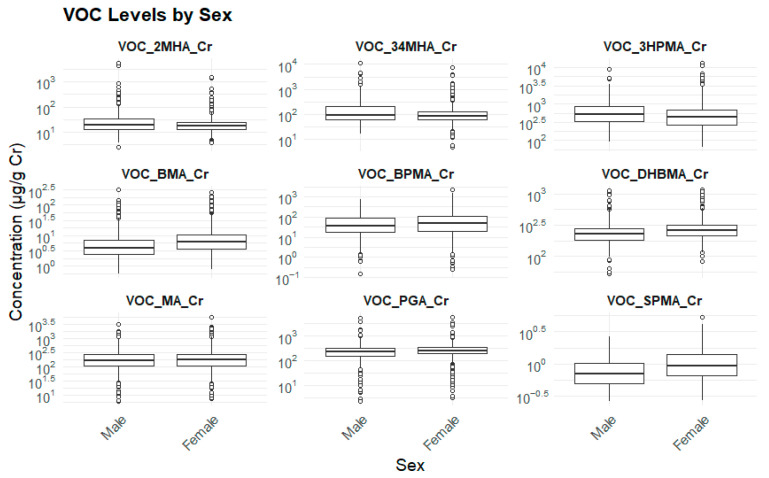
Sex and biomarker levels.

**Figure 3 toxics-13-00692-f003:**
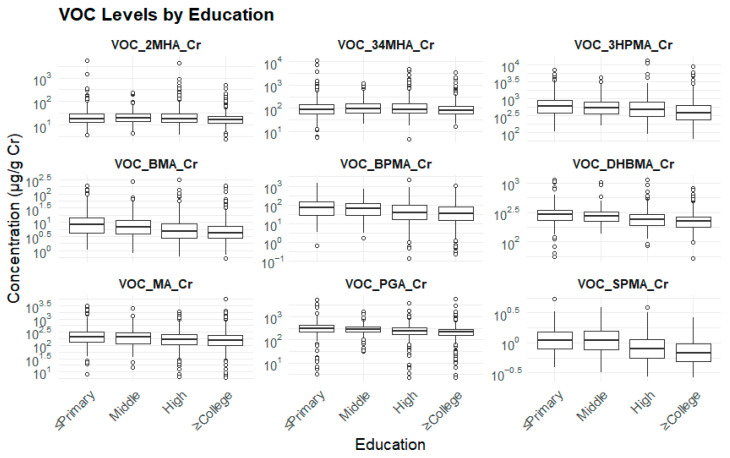
Education levels and biomarker levels.

**Figure 4 toxics-13-00692-f004:**
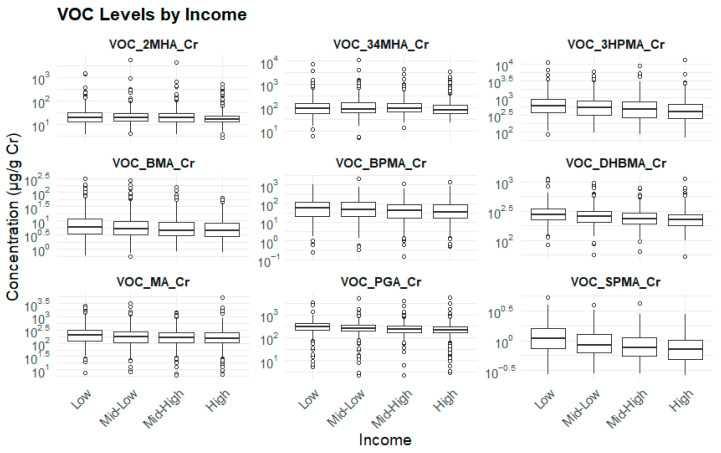
Household income levels and biomarker levels.

**Figure 5 toxics-13-00692-f005:**
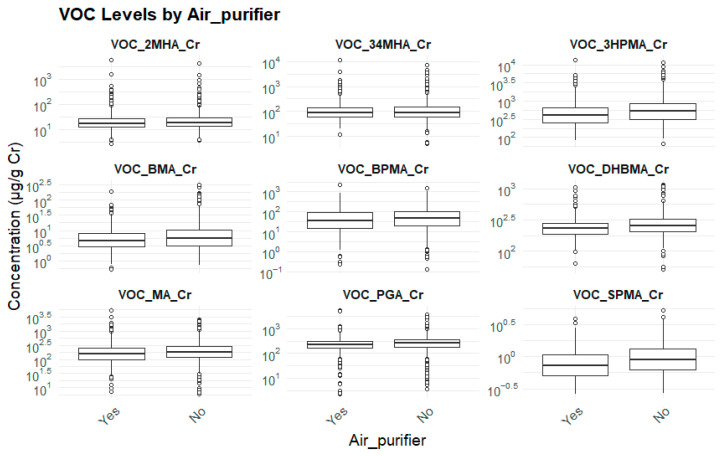
Usage of air purifier and biomarker levels.

**Figure 6 toxics-13-00692-f006:**
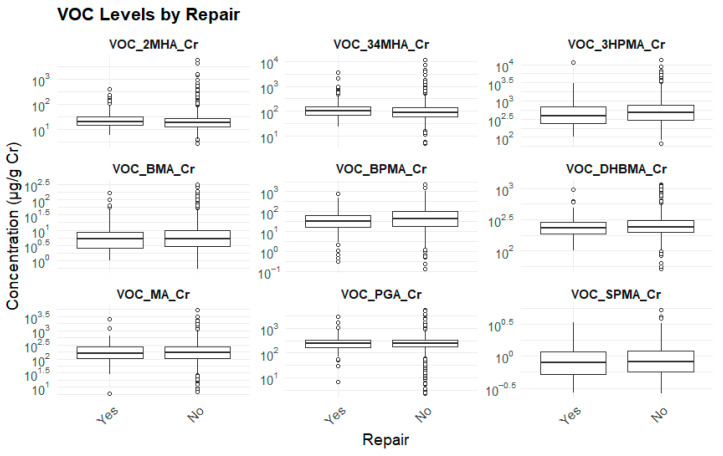
Home repairs and biomarker levels.

**Figure 7 toxics-13-00692-f007:**
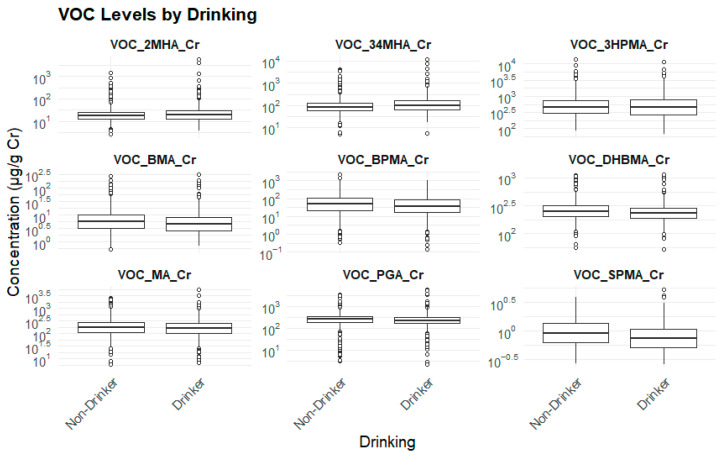
Drinking status and biomarker levels.

**Figure 8 toxics-13-00692-f008:**
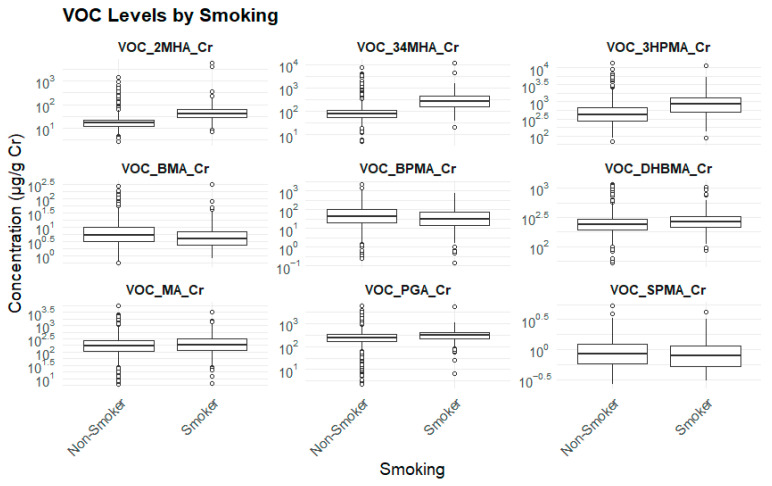
Smoking status and biomarker levels.

**Table 1 toxics-13-00692-t001:** VOC Biomarkers and LOD Values.

Parent Compound	Metabolite (Biomarker)	Abbreviation	LOD (μg/L)
Benzene	N-Acetyl-S-(phenyl)-L-cysteine	SPMA	0.251
Toluene	N-Acetyl-S-(benzyl)-L-cysteine	BMA	0.010
Ethylbenzene and Styrene	Phenylglyoxylic acid	PGA	0.985
Styrene	Mandelic acid	MA	5.337
Xylene	2-Methylhippuric acid	2-MHA	1.078
	3- and 4-Methylhippuric acid	3-MHA + 4-MHA	0.252
Acrolein	N-Acetyl-S-(3-hydroxypropyl)-L-cysteine	3-HPMA	0.128
1-Bromopropane	N-Acetyl-S-(n-propyl)-L-cysteine	BPMA	0.103
1,3-Butadiene	N-Acetyl-S-(3,4-dihydroxybutyl)-L-cysteine	DHBMA	0.179

**Table 2 toxics-13-00692-t002:** Sociodemographic characteristics.

Characteristic		Male 846 (45.3%)	Female 1034 (54.7%)	*p*-Value
Age (years), mean ± SE (IQR)		52.70 ± 0.786 (40–68)	53.35 ± 0.712 (40–69)	0.000 ***
Education, *n* (%)	≤Elementary School	98 (11.7)	254 (21.7)	0.000 ***
	Middle School	78 (7.7)	106 (11.1)	
	High School	306 (39.1)	287 (29.8)	
	≥College	364 (41.5)	387 (37.4)	
Household Income, *n* (%)	Low	151 (16.7)	229 (18.9)	0.200
	Low-Middle	219 (25.6)	278 (27.3)	
	Middle-High	234 (25.8)	259 (24.2)	
	High	287 (32.0)	316 (29.6)	
Drinking Status, *n* (%)	No	307 (33.1)	680 (61.7)	0.000 ***
	Yes	583 (66.9)	404 (38.3)	
Smoking Status, *n* (%)	No	634 (69.8)	1050 (96.7)	0.000 ***
	Yes	256 (30.2)	34 (3.3)	
Time at home on weekdays(hours), mean ± SE (IQR)		15.34 ± 0.228 (12–20)	17.87 ± 0.180 (14–22)	0.000 ***
Housing Repair within 6 months, *n* (%)	No	818 (91.4)	999 (91.1)	0.754
	Yes	75 (8.6)	88 (8.9)	
Air Purifier, *n* (%)	No	529 (58.1)	607 (54.7)	0.054
	Yes	364 (41.9)	480 (45.3)	

*** *p* < 0.001, IQR: interquartile range.

**Table 3 toxics-13-00692-t003:** Air pollutants and biomarker levels.

Variable	Mean ± SE	Variable	Mean ± SE
PM_2.5_ (μg/m^3^)	17.32 ± 0.758	2MHA (μg/g cr.)	36.75 ± 4.730
CO_2_ (ppm)	791.87 ± 17.983	3HPMA (μg/g cr.)	634.87 ± 21.211
HCHO (μg/ m^3^)	27.79 ± 0.981	BPMA (μg/g cr.)	79.70 ± 3.307
TVOC (μg/ m^3^)	257.26 ± 26.600	DHBMA (μg/g cr.)	263.06 ± 3.337
Benzene (μg/ m^3^)	5.21 ± 0.978	SPMA (μg/g cr.)	0.97 ± 0.020
Toluene (μg/ m^3^)	24.61 ± 2.338	BMA (μg/g cr.)	9.23 ± 0.433
Ethylbenzene (μg/ m^3^)	4.56 ± 0.393	PGA (μg/g cr.)	285.06 ± 7.698
Xylene (μg/ m^3^)	8.94 ± 0.909	3,4MHA (μg/g cr.)	172.74 ± 11.505
Styrene (μg/ m^3^)	3.73 ± 0.606	MA (μg/g cr.)	228.27 ± 9.164

**Table 4 toxics-13-00692-t004:** Sex and biomarker levels.

Biomarker (μg/g cr.)	Male (Mean ± SE)	Female (Mean ± SE)	*p*-Value
3HPMA	686.1 ± 55.6	723.5 ± 54.2	0.407
PGA	275.9 ± 33.6	308.2 ± 35.1	0.035 *
MA	212.3 ± 17.9	242.2 ± 21.9	0.038 *
SPMA	0.86 ± 0.04	1.10 ± 0.04	0.000 ***
BPMA	56.9 ± 11.8	78.3 ± 12.0	0.000 ***
DHBMA	250.2 ± 12.9	281.7 ± 11.9	0.000 ***
2MHA	55.38 ± 11.91	51.70 ± 10.46	0.359
BMA	5.05 ± 1.21	7.93 ± 1.27	0.000 ***
3,4MHA	258.9 ± 24.4	256.1 ± 23.9	0.888

* *p* < 0.05, *** *p* < 0.001.

**Table 5 toxics-13-00692-t005:** Education levels and biomarker levels.

Biomarker (μg/g cr.)	≤Elementary School ^a^ (Mean ± SE)	Middle School ^b^ (Mean ± SE)	High School ^c^ (Mean ± SE)	≥College ^d^ (Mean ± SE)	*p*-Value	Bonferroni
3HPMA	876.8 ± 67.1	698.9 ± 64.3	648.6 ± 53.9	618.9 ± 52.5	0.000 ***	^a^ > ^b^, ^a^ > ^c^, ^a^ > ^d^, ^b^ > ^d^
PGA	415.8 ± 23.4	335.2 ± 16.8	315.3 ± 15.8	281.1 ± 17.5	0.000 ***	^a^ > ^b^, ^a^ > ^c^, ^a^ > ^d^, ^b^ > ^c^, ^b^ > ^d^, ^c^ > ^d^
MA	273.4 ± 28.5	229.1 ± 21.7	196.6 ± 19.4	210.6 ± 24.1	0.008 **	^a^ > ^b^, ^a^ > ^c^, ^a^ > ^d^, ^b^ > ^c^, ^c^ > ^d^
SPMA	1.21 ± 0.041	1.26 ± 0.050	0.94 ± 0.031	0.84 ± 0.021	0.000 ***	^a^ > ^d^, ^a^ < ^b^, ^a^ > ^c^, ^b^ > ^c^, ^b^ > ^d^, ^c^ > ^d^
BPMA	98.7 ± 10.3	68.5 ± 12.2	57.4 ± 10.4	49.9 ± 9.5	0.000 ***	^a^ > ^b^, ^a^ > ^c^, ^a^ > ^d^, ^b^ > ^d^
DHBMA	289.9 ± 12.9	262.3 ± 16.1	264.3 ± 12.9	247.3 ± 13.1	0.001 **	^a^ > ^b^, ^a^ > ^c^, ^a^ > ^d^, ^b^ > ^d^, ^c^ > ^d^
2MHA	37.21 ± 6.13	40.87 ± 12.28	38.93 ± 9.27	25.19 ± 1.66	0.453	
BMA	9.76 ± 1.37	9.92 ± 3.21	4.42 ± 0.83	4.16 ± 0.84	0.000 ***	^a^ > ^c^, ^a^ > ^d^, ^b^ > ^c^, ^b^ > ^d^
3,4MHA	299.3 ± 72.4	231.8 ± 64.4	276.3 ± 65.1	219.4 ± 64.3	0.002 **	^a^ > ^b^, ^a^ > ^d^

** *p* < 0.01, *** *p* < 0.001, ^a^: ≤elementary school, ^b^: middle school, ^c^: high school, ^d^: ≥college.

**Table 6 toxics-13-00692-t006:** Household income levels and biomarker levels.

Biomarker (μg/g cr.)	Low Income ^a^ (Mean ± SE)	Low-Middle Income ^b^ (Mean ± SE)	Middle-High Income ^c^ (Mean ± SE)	High Income ^d^ (Mean ± SE)	*p*-Value	Bonferroni
3HPMA	733.8 ± 96.1	667.0 ± 59.0	656.3 ± 71.9	611.6 ± 57.7	0.042 *	^a^ > ^d^
PGA	327.6 ± 39.2	326.3 ± 19.9	320.3 ± 21.5	292.2 ± 18.4	0.316	
MA	223.8 ± 32.0	190.1 ± 24.1	206.3 ± 19.8	216.6 ± 19.7	0.666	
SPMA	1.20 ± 0.043	1.07 ± 0.032	1.02 ± 0.028	0.95 ± 0.028	0.000 ***	^a^ > ^b^, ^a^ > ^c^, ^a^ > ^d^, ^b^ > ^c^, ^b^ > ^d^, ^c^ > ^d^
BPMA	71.70 ± 14.39	77.05 ± 10.39	74.17 ± 9.89	62.21 ± 7.87	0.372	
DHBMA	291.8 ± 16.1	270.4 ± 12.0	267.4 ± 11.6	248.7 ± 11.6	0.003 *	^a^ > ^b^, ^a^ > ^c^, ^a^ > ^d^, ^b^ > ^d^, ^c^ > ^d^
2MHA	37.21 ± 6.13	40.87 ± 12.28	38.93 ± 9.27	25.19 ± 1.66	0.453	
BMA	7.14 ± 2.423	5.48 ± 1.264	7.26 ± 1.408	5.98 ± 1.490	0.684	
3,4MHA	317.2 ± 85.9	243.7 ± 26.5	307.4 ± 48.9	242.1 ± 41.2	0.460	

* *p* < 0.05, *** *p* < 0.001, ^a^: low income, ^b^: low-middle income, ^c^: middle-high income, ^d^: high income.

**Table 7 toxics-13-00692-t007:** Usage of air purifier and biomarker levels.

Biomarker (μg/g cr.)	Use (Mean ± SE)	No-Use (Mean ± SE)	*p*-Value
3HPMA	535.75 ± 23.68	714.87 ± 31.326	0.000 ***
PGA	264.02 ± 10.90	302.06 ± 10.59	0.014 *
MA	208.12 ± 11.98	244.53 ± 11.82	0.020 *
SPMA	0.86 ± 0.021	1.06 ± 0.028	0.000 ***
BPMA	72.94 ± 4.751	85.15 ± 4.227	0.045 *
DHBMA	242.16 ± 3.864	279.92 ± 4.289	0.000 ***
2MHA	55.72 ± 13.99	51.35 ± 10.04	0.359
BMA	7.13 ± 0.358	10.92 ± 0.739	0.000 ***
3,4MHA	170.98 ± 20.21	174.17 ± 12.939	0.895

* *p* < 0.05, *** *p* < 0.001.

**Table 8 toxics-13-00692-t008:** Home repairs and biomarker levels.

Biomarker (μg/g cr.)	No (Mean ± SE)	Yes (Mean ± SE)	*p*-Value
3HPMA	654.4 ± 58.8	767.2 ± 51.9	0.016 *
PGA	326.7 ± 18.0	345.2 ± 13.4	0.217
MA	226.6 ± 26.9	228.3 ± 16.7	0.949
SPMA	0.99 ± 0.06	1.03 ± 0.03	0.554
BPMA	62.1 ± 9.9	75.1 ± 9.3	0.076
DHBMA	262.9 ± 10.0	278.4 ± 8.9	0.043 *
2MHA	48.62 ± 8.76	58.46 ± 14.24	0.249
BMA	6.76 ± 0.91	7.36 ± 0.88	0.430
3,4MHA	247.4 ± 25.3	262.2 ± 26.6	0.558

* *p* < 0.05.

**Table 9 toxics-13-00692-t009:** Drinking status and biomarker levels.

Biomarker (μg/g cr.)	Non-Drinker (Mean ± SE)	Drinker (Mean ± SE)	*p*-Value
3HPMA	637.02 ± 25.950	631.14 ± 29.005	0.867
PGA	287.94 ± 9.231	282.69 ± 10.942	0.691
MA	239.77 ± 13.461	217.43 ± 10.743	0.163
SPMA	1.07 ± 0.030	0.88 ± 0.023	0.000 ***
BPMA	90.64 ± 5.127	69.47 ± 3.333	0.000 ***
DHBMA	275.58 ± 5.025	250.98 ± 3.879	0.000 ***
2MHA	52.06 ± 9.67	55.02 ± 12.71	0.541
BMA	10.53 ± 0.628	8.03 ± 0.464	0.000 ***
3,4MHA	143.84 ± 10.342	199.57 ± 18.171	0.004 **

** *p* < 0.01, *** *p* < 0.001.

**Table 10 toxics-13-00692-t010:** Smoking status and biomarker levels.

Biomarker (μg/g cr.)	Non-Smoker (Mean ± SE)	Smoker (Mean ± SE)	*p*-Value
3HPMA	510.7 ± 45.1	910.9 ± 68.1	0.000 ***
PGA	303.7 ± 12.7	368.2 ± 19.1	0.000 ***
MA	212.4 ± 16.9	242.5 ± 22.5	0.086
SPMA	1.00 ± 0.04	1.02 ± 0.04	0.639
BPMA	74.7 ± 8.6	62.3 ± 10.2	0.050
DHBMA	251.0 ± 7.7	291.1 ± 11.8	0.000 ***
2MHA	23.55 ± 3.62	83.53 ± 24.02	0.024 *
BMA	7.73 ± 0.85	6.90 ± 0.91	0.280
3,4MHA	148.97 ± 21.5	368.83 ± 32.09	0.000 ***

* *p* < 0.05, *** *p* < 0.001.

**Table 11 toxics-13-00692-t011:** Age and biomarker levels.

Biomarker (μg/g cr.)	Estimate	Std. Error	*p*-Value
3HPMA	−4.4015 × 10^−4^	0.001	0.965
PGA	0.007	0.003	0.007 **
MA	−0.001	0.002	0.781
SPMA	9.377	1.062	0.000 ***
BPMA	0.008	0.005	0.086
DHBMA	0.025	0.005	0.000 ***
2MHA	0.009	0.005	0.077
BMA	0.103	0.023	0.000 ***
3,4MHA	−0.007	0.003	0.007 **

** *p* < 0.01, *** *p* < 0.001.

**Table 12 toxics-13-00692-t012:** Time at home on weekdays (hours per day) and biomarker levels.

Biomarker (μg/g cr.)	Estimate	Std. Error	*p*-Value
3HPMA	0.000	0.001	0.654
PGA	0.001	0.001	0.080
MA	−3.014 × 10^−5^	0.001	0.965
SPMA	1.574	0.272	0.000 ***
BPMA	0.002	0.001	0.089
DHBMA	0.005	0.001	0.000 ***
2MHA	0.003	0.002	0.184
BMA	0.007	0.006	0.235
3,4MHA	−0.002	0.001	0.012 *

* *p* < 0.05, *** *p* < 0.001.

**Table 13 toxics-13-00692-t013:** Multilinear regression analysis results for biomarkers.

Biomarker	Variable	Estimate	Std. Error	*p*-Value
3HPMA (μg/g cr.) Adj. R^2^ = 0.157	Age	4.684	1.616	0.005 **
	Smoking	402.342	89.940	0.000 ***
	Air purifier	−136.877	54.355	0.013 *
	Ethylbenzene	2.044	0.626	0.001 **
	Styrene	5.154	2.310	0.028 *
PGA (μg/g cr.) Adj. R^2^ = 0.481	Age	2.384	0.610	0.000 ***
	Sex	−35.961	16.605	0.033 *
	Smoking	83.698	24.180	0.001 **
	BMI	−4.915	1.982	0.015 *
	Air Purifier	−56.536	15.038	0.001 **
	CO_2_	−0.065	0.027	0.016 *
	Benzene	6.317	0.972	0.000 ***
	Ethylbenzene	1.837	0.553	0.001 **
MA (μg/g cr.) Adj. R^2^ = 0.050	Age	1.390	0.634	0.031 *
2MHA (μg/g cr.) Adj. R^2^ = 0.058	Benzene	0.841	0.245	0.001 **
SPMA (μg/g cr.) Adj. R^2^ = 0.246	Age	0.008	0.002	0.000 ***
	Sex	−0.221	0.048	0.000 ***
	Education	−0.200	0.063	0.016 *
	Air purifier	−0.180	0.054	0.001 **
	Xylene	−0.001	0.000	0.020 *
BPMA (μg/g cr.) Adj. R^2^ = 0.088	Age	0.904	0.317	0.000 ***
	Smoking	18.223	8.383	0.032 *
	CO_2_	−0.021	0.010	0.034 *
	Ethylbenzene	0.468	0.089	0.000 ***
DHBMA (μg/g cr.) Adj. R^2^ = 0.210	Age	1.230	0.486	0.013 *
	Sex	−47.791	8.288	0.000 ***
	Education	−23.509	9.662	0.016 *
	Income	−18.677	7.773	0.018 *
	Smoking	56.789	14.455	0.000 ***
	Air purifier	−20.787	9.780	0.032 *
	CO_2_	−0.029	0.012	0.019 *
	Xylene	0.307	0.136	0.029 *
BMA (μg/g cr.) Adj. R^2^ = 0.108	Age	0.171	0.047	0.000 ***
	Drinking	−3.398	1.113	0.003 **
	Smoking	2.714	1.067	0.012 *
3.4MHA (μg/g cr.) Adj. R^2^ = 0.171	Smoking	270.180	70.437	0.000 ***
	BMI	−8.545	4.148	0.044 *
	HCHO	1.520	0.660	0.023 *
	Benzene	3.378	1.029	0.000 ***
	Ethylbenzene	2.910	0.784	0.000 ***

* *p* < 0.05, ** *p* < 0.01, *** *p* < 0.001, Adjusted for age, sex, education, household income, house repair within 6 months, using air purifier, alcohol consumption, smoking status, BMI, home staying hours, PM_2.5_, CO_2_, HCHO, TVOC, Benzene, Toluene, Ethylbenzene, Xylene, Styrene.

**Table 14 toxics-13-00692-t014:** Pearson’s correlation between indoor air pollutants and urinary metabolites.

	PM_2.5_	CO_2_	HCHO	TVOC	Benzene	Toluene	Ethylbenzene	Xylene	Styrene
BMA	−0.006	−0.059 *	−0.0569	−0.006	0.003	0.008	0.004	0.023	0.026
2MHA	0.003	0.078 **	0.004	0.047 *	0.076 *	0.029	0.061 *	0.079 **	0.023
3,4MHA	0.040	0.051 *	0.012	0.090 **	0.151 **	0.092 **	0.205 **	0.211 **	0.069 *
PGA	0.017	0.004	0.039	0.126 **	0.302 **	0.123 **	0.245 **	0.277 **	0.085 *
MA	0.044	0.028	−0.005	0.022	0.010	−0.020	0.039	0.039	−0.005
SPMA	0.042	−0.049 *	−0.083 **	0.005	−0.039	−0.050 *	0.007	−0.001	0.015
3HPMA	0.078 **	−0.019	−0.053 *	0.013	−0.022	−0.030	0.041	0.029	0.054
BPMA	−0.031	−0.036	−0.008	0.003	−0.027	0.017	0.086 **	0.056 *	−0.028
DHBMA	0.080 **	−0.089 **	−0.066 **	0.009	0.006	−0.020	0.023	0.060 *	0.004

* *p* < 0.05, ** *p* < 0.01.

**Table 15 toxics-13-00692-t015:** Pearson’s correlation between indoor air pollutants.

	PM_2.5_	CO_2_	HCHO	TVOC	Benzene	Toluene	Ethylbenzene	Xylene	Styrene
PM_2.5_									
CO_2_	0.014								
HCHO	−0.040	0.450 **							
TVOC	0.034	0.325 **	0.241 **						
Benzene	−0.043	0.161 **	0.084 *	0.446 **					
Toluene	−0.002	0.201 **	0.175 **	0.420 **	0.490 **				
Ethylbenzene	0.004	0.065 **	0.102 **	0.242 **	0.143 **	0.240 **			
Xylene	0.002	0.084 **	0.064 *	0.374 **	0.448 **	0.306 **	0.748 **		
Styrene	0.030	0.171 **	0.146 **	0.480 **	0.185 **	0.196 **	0.151 **	0.374 **	

* *p* < 0.05, ** *p* < 0.01.

## Data Availability

The data that support the findings of this study are openly available in Korea Centers for Disease Control and Prevention National Health and Nutrition Survey Data Sharing Service at https://knhanes.kdca.go.kr/knhanes/dataAnlsGd/utztnGd.do;jsessionid=3hVBHDpLApT2bsMagAXCNWWI-dJvpl9fL-684t6X.knhanes_10 (accessed on 10 August 2025).

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
