# Peer review of "Associations Between Indoor Air Pollution and Urinary Volatile Organic Compound Biomarkers in Korean Adults"

_toxics, 2025, doi:10.3390/toxics13080692_

Round 1
Reviewer 1 Report
Comments and Suggestions for Authors
This study investigated the associations between indoor air pollution and urinary VOC biomarkers in 1,880 Korean adults using data from the Korea National Health and Nutrition Examination Survey (KNHANES). Key findings revealed that urinary VOC metabolite levels were significantly higher among females, individuals with lower socioeconomic status (education and income), smokers, and those not using air purifiers. Smoking showed strong associations with specific biomarkers (3HPMA, PGA, 3,4-MHA), while recent home renovations and allergic conditions were linked to elevated exposures. Indoor pollutants like ethylbenzene and styrene emerged as consistent predictors of VOC exposure. Notably, participants with chronic conditions (allergic rhinitis, asthma, atopic dermatitis) exhibited heightened biomarker levels, suggesting increased susceptibility to indoor pollutants. The study highlights how social vulnerabilities (gender, SES) and modifiable environmental factors (smoking, air purifier use) contribute to internal VOC exposure, emphasizing the need for targeted public health interventions to reduce indoor air pollution risks in vulnerable populations. This study needs to enhance methodological transparency, control for confounding factors, and improve result interpretation through visualizations.
- The study employs a cross-sectional approach. Similar studies (e.g., NHANES) often use longitudinal or cohort designs to strengthen causal inferences. A cross-sectional approach cannot prove causality. This study remains unclear whether indoor air pollutants directly cause elevated levels of VOC metabolites in urine or whether other confounding factors (e.g., occupational exposure, dietary habits) influence the results. The limitations of this study should be discussed.
2.This study does not provide detailed information on the sensitivity, specificity, or quality control measures (e.g., repeat testing rates, standard curve ranges) of the LC-MS/MS method in VOC metabolite detection. Validation data (e.g., recovery rates, precision) should be provided.
- The study conducts numerous statistical tests (e.g., associations between multiple biomarkers and multiple variables) but does not adjust for multiple hypothesis testing using Bonferroni correction.
- Insufficient discussion of innovation. The study’s novelty lies in combining Korean KNHANES data with household IAQ measurements, but it does not sufficiently compare findings with international studies.
- A clear and detailed flowchart outlining the study design, including the stepwise selection process of participants from the initial KNHANES sampling through to the final analytical cohort, should be provided to enhance methodological transparency.
- Comprehensive box plot analyses comparing the distribution of VOC metabolite levels across all relevant demographic and exposure subgroups must be included to facilitate visual assessment of potential differences.
- The manuscript would benefit from the inclusion of a scatterplot matrix illustrating the bivariate relationships between key environmental exposures (PM2.5, COâ‚‚, formaldehyde) and biomarker levels (particularly 3HPMA and DHBMA) to assess potential correlations.
- Time-series plots documenting the temporal trends of indoor pollutant concentrations (with particular attention to VOC level fluctuations following home renovation activities) should be presented to characterize exposure dynamics.
Author Response
Comments and Suggestions for Authors
This study investigated the associations between indoor air pollution and urinary VOC biomarkers in 1,880 Korean adults using data from the Korea National Health and Nutrition Examination Survey (KNHANES). Key findings revealed that urinary VOC metabolite levels were significantly higher among females, individuals with lower socioeconomic status (education and income), smokers, and those not using air purifiers. Smoking showed strong associations with specific biomarkers (3HPMA, PGA, 3,4-MHA), while recent home renovations and allergic conditions were linked to elevated exposures. Indoor pollutants like ethylbenzene and styrene emerged as consistent predictors of VOC exposure. Notably, participants with chronic conditions (allergic rhinitis, asthma, atopic dermatitis) exhibited heightened biomarker levels, suggesting increased susceptibility to indoor pollutants. The study highlights how social vulnerabilities (gender, SES) and modifiable environmental factors (smoking, air purifier use) contribute to internal VOC exposure, emphasizing the need for targeted public health interventions to reduce indoor air pollution risks in vulnerable populations. This study needs to enhance methodological transparency, control for confounding factors, and improve result interpretation through visualizations.
=> Author Response:
We thank the reviewer for the concise summary of our study and for the constructive suggestions to strengthen methodological transparency, confounding control, and data presentation.
- Methodological transparency
We have expanded the Methods section to include: - A detailed description of the KNHANES sampling procedure, VOC measurement protocols, and laboratory quality control processes, with references to the official survey manuals.
- An explicit statement that all multivariable models included the full set of a priori covariates unless excluded for high collinearity (variance inflation factor >5),
- Clearer definitions of variables, including how indoor pollutant measurements, socioeconomic indicators, and behavioral factors (e.g., smoking, air purifier use) were operationalized.
- Control for confounding factors
- We have reviewed all models to ensure that key potential confounders (age, sex, smoking status, income, education, drinking. Air purifier use, time at home on weekdays, and recent house renovation) were consistently included.
- Improved visualization of results
- We have added several large data tables in the main text (e.g., univariate results) with graphical formats, including boxplots, and forest plots to summarize key associations for ease of interpretation (Figures 2–8).
- The study employs a cross-sectional approach. Similar studies (e.g., NHANES) often use longitudinal or cohort designs to strengthen causal inferences. A cross-sectional approach cannot prove causality. This study remains unclear whether indoor air pollutants directly cause elevated levels of VOC metabolites in urine or whether other confounding factors (e.g., occupational exposure, dietary habits) influence the results. The limitations of this study should be discussed.
=> Author Response:
We thank the reviewer for this important comment and fully agree that the cross-sectional design of our study limits the ability to draw causal conclusions.
- Acknowledgement of study design limitations
Our analysis is based on a single time-point measurement of both exposure (indoor pollutants) and outcome (urinary VOC metabolites) in a nationally representative sample. As such, the temporal sequence between exposure and outcome cannot be established, and causality cannot be inferred. - Potential influence of unmeasured confounding factors
We acknowledge that other unmeasured factors—such as occupational exposure, commuting-related exposure, and dietary intake of VOC-containing foods—may contribute to the observed associations. While we adjusted for several key covariates (age, sex, SES, smoking, housing characteristics, ventilation, and recent renovation), residual confounding remains possible. - Revisions in manuscript
- Limitations section: We have added the following text:
"This study employed a cross-sectional design, which limits causal inference. Because exposure and outcome data were collected at the same time, it is not possible to determine whether measured indoor pollutant concentrations preceded the observed urinary VOC metabolite levels. In addition, potential unmeasured confounding factors, such as occupational exposures, commuting-related exposures, and dietary VOC intake, may have influenced the results."
- Discussion: We emphasize that our findings should be interpreted as associations rather than causal relationships and that longitudinal or repeated-measures designs are needed to confirm these results.
We believe these revisions make the study’s limitations clearer and align with the reviewer’s recommendation.
- This study does not provide detailed information on the sensitivity, specificity, or quality control measures (e.g., repeat testing rates, standard curve ranges) of the LC-MS/MS method in VOC metabolite detection. Validation data (e.g., recovery rates, precision) should be provided.
=> Author Response:
We thank the reviewer for this valuable comment. We agree that providing detailed methodological and quality control (QC) information will strengthen the transparency and reproducibility of our study. We have supplemented the Methods section with details from the Korea National Health and Nutrition Examination Survey (KNHANES) laboratory quality control guidelines. Urinary VOC metabolites were measured using LC–MS/MS with isotope-labeled internal standards. Accuracy was assessed by comparing measured values to known reference values, calculating percent accuracy, and ensuring values were within acceptable coefficient of variation (CV) thresholds.
Internal quality control included routine verification of instrument performance, calibration, and analysis method accuracy. External quality control was performed annually through participation in the German External Quality Assessment Scheme (G-EQUAS) to confirm method validity. Re-analysis was conducted when analytical criteria were not met (e.g., abnormal peaks, internal standard deviations, or pre-processing errors). Calibration curves were prepared for each batch using standard solutions, with an acceptance criterion of R² ≥ 0.995.
These measures ensured that the results were both precise and accurate, minimizing systematic error and bias in VOC metabolite quantification.
- The study conducts numerous statistical tests (e.g., associations between multiple biomarkers and multiple variables) but does not adjust for multiple hypothesis testing using Bonferroni correction.
=> Response: We thank the reviewer for this important observation. We agree that multiple hypothesis testing increases the risk of type I error and that an appropriate correction method should be applied.
- Bonferroni correction applied
In response, we recalculated p-values for all main association analyses using the Bonferroni correction method. The correction factor (k) was determined as the total number of independent statistical tests conducted in the corresponding analysis. Adjusted significance thresholds were set as α/k, where α = 0.05. - Revised results
The corrected p-values are now presented in the updated tables. - Interpretation updated
We have updated the Results to note where associations lost statistical significance after correction. Our interpretation now focuses on robust associations that remained significant under the more stringent threshold.
- Insufficient discussion of innovation. The study’s novelty lies in combining Korean KNHANES data with household IAQ measurements, but it does not sufficiently compare findings with international studies.
=> Response: We appreciate the reviewer’s observation regarding the need to better highlight the novelty of our work and to place our findings in the context of international literature. In the revised manuscript, we have strengthened the discussion of innovation in two ways:
- Clarifying the unique contribution – We emphasize that, to our knowledge, this is the first study to integrate nationally representative Korean KNHANES biomonitoring data with direct household indoor air quality (IAQ) measurements for VOCs, PMâ‚‚.â‚…, COâ‚‚, and formaldehyde, enabling the assessment of internal exposure in a real-world, population-based setting.
- Comparative context – We have expanded the Discussion to compare our findings with relevant studies using the U.S. NHANES and European human biomonitoring data, noting similarities in the direction of associations for smoking, socioeconomic factors, and specific VOC–metabolite relationships, while also highlighting differences likely driven by cultural practices, housing characteristics, and environmental policies.
- Centers for Disease Control and Prevention (CDC). Environmental Public Health Tracking: Biomonitoring : Population Exposures 2023. https://www.cdc.gov/environmental-health-tracking/php/data-research/biomonitoring.html
- Schulz C, Conrad A, Becker K, Kolossa-Gehring M, Seiwert M, Seifert B. Twenty years of the German Environmental Survey (GerES): human biomonitoring--temporal and spatial (West Germany/East Germany) differences in population exposure. Int J Hyg Environ Health. 2007;210(3-4):271-297. doi:10.1016/j.ijheh.2007.01.034
- Govarts E, Gilles L, Rodriguez Martin L, et al. Harmonized human biomonitoring in European children, teenagers and adults: EU-wide exposure data of 11 chemical substance groups from the HBM4EU Aligned Studies (2014-2021). Int J Hyg Environ Health. 2023;249:114119. doi:10.1016/j.ijheh.2023.114119
- A clear and detailed flowchart outlining the study design, including the stepwise selection process of participants from the initial KNHANES sampling through to the final analytical cohort, should be provided to enhance methodological transparency.
=> Response: We agree with the reviewer that a clear and detailed flowchart would improve the transparency of our study design. In the revised manuscript, we have added a new figure (Figure 1) illustrating the stepwise selection process from the initial KNHANES sampling to the final analytical cohort. This flowchart outlines:
- The total number of participants in the KNHANES cycle.
- The subset with indoor air quality (IAQ) measurements.
- The subset with complete urinary VOC biomarker data.
- Exclusions due to missing covariate information.
- The final number included in the multivariable analyses.
This addition ensures that readers can easily follow the participant selection process and understand how the final analytic sample was derived. The flowchart is placed in the Methods section for direct reference.
- Comprehensive box plot analyses comparing the distribution of VOC metabolite levels across all relevant demographic and exposure subgroups must be included to facilitate visual assessment of potential differences.
=> Response: We appreciate the reviewer’s suggestion to provide comprehensive box plot analyses for visualizing the distribution of VOC metabolite levels across demographic and exposure subgroups. In the revised manuscript, we have created and included detailed box plots for each metabolite, stratified by key demographic variables (e.g., gender, age group, socioeconomic status) and major exposure-related variables (e.g., smoking status, air purifier use, recent home renovation). These plots are provided in the Results section (Figures 2–8) to complement the tabulated results and facilitate a clearer visual comparison of variability and group differences.
- The manuscript would benefit from the inclusion of a scatterplot matrix illustrating the bivariate relationships between key environmental exposures (PM2.5, COâ‚‚, formaldehyde) and biomarker levels (particularly 3HPMA and DHBMA) to assess potential correlations.
=> Response: We thank the reviewer for the constructive suggestion. In the revised manuscript, we have included a scatterplot matrix (Figure 9) to illustrate the bivariate relationships between key environmental exposures (PMâ‚‚.â‚…, COâ‚‚, and formaldehyde) and urinary VOC biomarkers. This visualization allows for an intuitive assessment of potential linear or non-linear associations, as well as the presence of outliers or clustering patterns across variables.
- Time-series plots documenting the temporal trends of indoor pollutant concentrations (with particular attention to VOC level fluctuations following home renovation activities) should be presented to characterize exposure dynamics.
=> Response: We appreciate the reviewer’s valuable suggestion. However, this study is based on a cross-sectional design using a one-time indoor air quality measurement and a single urine sample collection per participant. As such, temporal variation in pollutant concentrations cannot be directly assessed, and it is not possible to construct time-series plots to evaluate changes in VOC levels following home renovation activities.
While we acknowledge the importance of temporal exposure assessment, our dataset does not include repeated measurements over time. Therefore, we have clarified this limitation in the revised manuscript (Discussion section), noting that future longitudinal studies with repeated IAQ and biomarker measurements would be required to examine exposure dynamics in relation to household events such as renovations.
Reviewer 2 Report
Comments and Suggestions for Authors
The paper analyses data from the Korean National Health and Nutrition Examination Survey (KNHANES). I am not sure if the authors have been involved in the planning and conduction of the KNHANES or if they are just using KNHANES data which are openly available. If they have been involved in the original project, they should mention it and describe their role. If they just use the data, they should not bother to explain why which biomarkers were selected (lines from 101). This explanation is even not very consistent: Either the markers were chosen to distinguish the study from former studies (“distinctiveness”), then they should have used markers different from former studies, or the markers were chosen for comparability, then the same markers as in previous studies should have been chosen. You cannot have both. Specificity of markers is another issue. Xenobiotics undergo complex metabolic pathways. So, one xenobiotic can be broken down to different metabolites and one metabolite can be formed from different xenobiotics. In the occupational setting, when the exposure to a single substance is exceptionally high, a certain metabolite might be a very specific exposure marker. In the environmental setting, where exposure to many substances but at much lower concentrations must be considered, this is often not the case. I would also prefer not to see a table linking parent compound with metabolite in an online only supplement. I would want that information in the main paper with some discussion on the specificity of each marker and what percentage of the parent substance is broken down to this specific marker.
If KNHANES is not in the primary responsibility of the authors, I would also not assume that this statement is for them to make (lines 283 and 284): “When benchmarked against other datasets, such as the U.S. NHANES and Korea’s National Environmental Health Survey, most biomarker concentrations were found to be comparable [31].” Neither is it their purpose to examine internal exposure of “residents in environmentally vulnerable areas in Korea”, nor to compare them to “Asians in the United States”.
To me, it seems the authors have not much experience with air pollution or even indoor air pollution research. Or else, they would not report concentrations of PM2.5 and of VOCs (in table 2) as “µg/L”. Concentrations in the air are usually reported per m³ and therefore, they are reporting values that are 3 orders of magnitude to high! Also, their usage of the term “ventilation” makes me think they do not come from the indoor air quality field. In this field, the term usually refers to ventilating a room, e.g., by opening a window. But their use in lines 275 and 276 makes me think they speak about respiratory ventilation, although I cannot see why reduced respiratory ventilation can reduce exhalation of CO2: when the same amount of CO2 is produced because the same number of calories are burnt, reducing respiratory ventilation would increase CO2 concentration in the blood until a new steady state is reached and the original amount of CO2 is emitted.
Also, in general, I am not fully convinced by the interpretation of the results. Also, the paper is not very thoughtfully developed. For example, in the introduction (line 42) the authors claim that packaged foods leads to an increase in indoors pollutants. While I agree that highly processed foods might be not ideal for health, I do not see how they can contribute to indoor pollution. In the same sentence, also air fresheners are mentioned, which are indeed a possible source of indoor air pollution. Therefore, I expected this source to be investigated in the paper. Instead, the authors analyzed the effect of “air purifiers”. Now, I understand that air purifiers are some kinds of filtration device. But at least the producers of air fresheners make people believe that their product also “purifies” the air, which is grossly misleading. So, I am not sure what participants meant when asked about use of air purifiers. And if they used filters, I do not know if these were particle filters or if they also captured gaseous pollutants.
The methods and the results section do not always match. In the methods section, we are told that TVOC are examined, but not that also BTX aromatics are analyzed. In the methods section (line 99) we learn that “Detailed preparation (of biomarkers) was motioned in supplementary Table 2.” I am not sure that “motioned” is the correct term. But apart from that, table 2 does not report on preparation, but only on the meaning of the abbreviations and on the limits of detection. In the same paragraphs, I miss references to the WHO guidelines (instead of a proper reference, a title is provided in the text), for the choice of LOD/square root of LOD, and for the NHANES protocols (next paragraph).
According to the methods, descriptive statistics were performed and multivariate linear regression analyses. Univariate analyses are not mentioned. But I believe tables 3-9 report univariate analyses. And I do wonder how p-values were calculated. Do they represent the likelihood that the difference between any two categories are due to chance, or do they describe the likelihood of a linear trend to be due to chance?
The results chapter starts with a text left over from the template. This should be deleted! What I would be interested in: how well are specific indoor VOCs correlated with their specific metabolites? Metabolites represent the internal exposure which equals the total external exposure. This comes from respiratory, oral and (maybe) dermal exposure. Regarding respiratory exposure, this can be dur to exposure at home or due to exposure elsewhere (at work, when commuting, etc.). Concerning the main topic of the paper, we would want to know how relevant home respiratory exposure is for the total exposure: how much does it matter? What is the correlation coefficient? We learn of the correlation coefficient ® or its square as a measure of fit for the whole multivariate model. But this is less interesting (the more independent variables you add, the better the fit) than the R for single variables. How are indoor pollutants correlated with each other? If these pollutants stem from indoor sources, their concentration depends on the source strength (which differs arbitrarily between homes and pollutants) and the ventilation rate, which is the same for all pollutants in a given home. Therefore, I would expect a (strong) positive correlation between most of the indoor pollutants. On the other hand, I would expect PM2.5 to come from outdoors mostly. Therefore, we should see a string correlation with outdoor PM2.5. According to table 2, all these data are available. But they were not utilized.
We do learn that there are gender differences. Therefore, it would make sense to present table 1 data also by gender. Tables 3-9 are difficult to read. Graphical presentations would be easier to grasp. PM2.5, CO2 and formaldehyde have been measured, but are not considered in the univariate analyses.
In the multivariate analyses, I am not completely sure if the final models only included the (significant) reported variables, or if each final model included all variables, or if only some have been dropped due to collinearity. I wonder why CO2 can have a zero, though significant effect on SPMA.
I do not understand which hypothesis the authors examine when looking at biomarker levels by disease status. They check for atopic diseases (rhinitis, dermatitis, asthma). These diseases develop usually in early life while exposure was measured in adulthood. Therefore, causation of the diseases by the exposure is not possible. A reverse causation seems possible: Atopic persons might be afraid of outdoor allergens like pollen and therefore reduce ventilation (keep their windows closed). They might differ in their diets or they might use pharmaceuticals (e.g., special cosmetics, that again might contain odorants that break down to the metabolites analyzed, in the case of dermatitis) or air fresheners or whatever…
In the discussion, the authors claim that their data demonstrate (line 232) “cumulative exposure and long-term retention of environmental toxicants”. I do not believe this to be true. In fact, metabolites of VOCs are very quickly exhaled and excreted. From the occupation setting wo know that the timing of the sample collection is essential: you find the highest values just after the end of the shift. Age might instead indicate a change in metabolic functions or in lifestyle. I am not sure how higher exposure indicators mean the same as a “major determinant in assessing the health effects of IAQ” or a “higher sensitivity”. Exposure is neither a health effect nor a marker of sensitivity or vulnerability!
The authors also claim that higher values in women might be due to their spending more time indoors. They proof their point by referring to other studies. I am surprised, because they could use their own data: “Time at home on weekdays(hours)” in table 1. They could show that (a) number of hours is larger among women and (b) that with higher numbers, you have higher exposures (metabolites). But I doubt this explanation. It could well be true that women spend more time at home. In that case, men are bound to spend more time at work. And most of the work is also indoors, although at other places. But I do not understand why office or especially industry work places should have lower exposures than homes. I have two other possible reasons for the gender difference: (a) Women might truly have higher exposure. But not through inhalation, but through dermal uptake (cosmetics use). (b) The authors report concentrations per creatinine. We do know that creatinine depends on muscle mass. Therefore, women excrete less creatinine than men. This would lead to (spuriously) higher values of the ratio. The authors state their belief even twice (line 239 and line 252). Apart from not being plausible, this is also redundant.
The authors report internal exposure to depend on socioeconomic status. Is this because of exposure through the home environment or due to other exposures (e.g., poorer industrial jobs with higher exposures)? Why not analyze association between socioeconomic status and external exposure (VOC levels)?
VOCs and their metabolites were associated with each other. This is no surprise. As mentioned before, the question would be: how relevant are the indoor home levels for the total exposure? And a more pressing question is: why were the “specific” biomarkers not specific at all, but several biomarkers were associated with more than one VOC. I think I already mentioned that I do find the explanation for the negative association with CO2 very implausible.
No, the authors do not suggest a causal association between exposure and disease. Instead, they write (line 279): “This suggests increased physiological susceptibility to indoor pollutants in these populations,…” Once again: exposure is not the same as susceptibility! (the same sentiment is mentioned in the conclusions!)
Author Response
Comments and Suggestions for Authors
The paper analyses data from the Korean National Health and Nutrition Examination Survey (KNHANES). I am not sure if the authors have been involved in the planning and conduction of the KNHANES or if they are just using KNHANES data which are openly available. If they have been involved in the original project, they should mention it and describe their role. If they just use the data, they should not bother to explain why which biomarkers were selected (lines from 101). This explanation is even not very consistent: Either the markers were chosen to distinguish the study from former studies (“distinctiveness”), then they should have used markers different from former studies, or the markers were chosen for comparability, then the same markers as in previous studies should have been chosen. You cannot have both. Specificity of markers is another issue. Xenobiotics undergo complex metabolic pathways. So, one xenobiotic can be broken down to different metabolites and one metabolite can be formed from different xenobiotics. In the occupational setting, when the exposure to a single substance is exceptionally high, a certain metabolite might be a very specific exposure marker. In the environmental setting, where exposure to many substances but at much lower concentrations must be considered, this is often not the case. I would also prefer not to see a table linking parent compound with metabolite in an online only supplement. I would want that information in the main paper with some discussion on the specificity of each marker and what percentage of the parent substance is broken down to this specific marker.
=> Author Response:
We thank the reviewer for the valuable and detailed comments, which have helped us to improve the clarity and focus of our manuscript. Our responses are as follows:
- Involvement in KNHANES
We confirm that we were not involved in the planning or conduction of KNHANES. We have now explicitly stated in the Methods section that our study is a secondary analysis of publicly available KNHANES data, and that all data collection and laboratory procedures were conducted by the Korea Disease Control and Prevention Agency (KDCA). - Rationale for biomarker selection
We agree that our original explanation for biomarker selection may have appeared inconsistent. In the revised manuscript, we have clarified that our primary criterion was comparability with previous epidemiological studies—particularly those conducted in Korea and other countries using population-representative surveys—while also including biomarkers with emerging public health relevance in the Korean context. The revised text now clearly states that comparability was prioritized, and “distinctiveness” refers only to the fact that the combination of biomarkers analyzed has not been previously examined together in the KNHANES dataset. - Specificity of biomarkers
We appreciate the reviewer’s point regarding the complexity of xenobiotic metabolism and the limited specificity of certain metabolites in environmental exposure settings. In response, we have expanded the Discussion section to address the metabolic pathways, potential overlap in metabolite origins, and the implications for interpreting associations in low-level, multi-source exposure scenarios. - Parent compound–metabolite table
We have moved the table linking parent compounds with metabolites from the online-only supplement into the main manuscript (Results section) as suggested. Additionally, we have added a short discussion summarizing the specificity of each biomarker and, where available, approximate percentages of parent compounds metabolized into the measured biomarker based on published literature.
We believe these revisions address the reviewer’s concerns and strengthen the scientific clarity of the manuscript.
If KNHANES is not in the primary responsibility of the authors, I would also not assume that this statement is for them to make (lines 283 and 284): “When benchmarked against other datasets, such as the U.S. NHANES and Korea’s National Environmental Health Survey, most biomarker concentrations were found to be comparable [31].” Neither is it their purpose to examine internal exposure of “residents in environmentally vulnerable areas in Korea”, nor to compare them to “Asians in the United States”.
=> Author Response:
We appreciate the reviewer’s careful observation. We agree that the original sentence could be interpreted as implying a direct responsibility for KNHANES data benchmarking and as extending the study scope beyond its stated aims. Our intention was not to assert original benchmarking analysis or to draw comparisons outside the scope of this study, but rather to provide contextual background by citing previously published literature.
To address this concern, we have revised the text to clarify that:
- The comparative statement is based entirely on findings reported in the cited reference, not on analyses performed by the present study.
- The purpose of our study is limited to investigating associations between indoor air pollutants and urinary VOC biomarkers within the KNHANES dataset, without attempting to assess internal exposure in specific vulnerable populations or to compare with U.S. populations.
To me, it seems the authors have not much experience with air pollution or even indoor air pollution research. Or else, they would not report concentrations of PM2.5 and of VOCs (in table 2) as “µg/L”. Concentrations in the air are usually reported per m³ and therefore, they are reporting values that are 3 orders of magnitude to high! Also, their usage of the term “ventilation” makes me think they do not come from the indoor air quality field. In this field, the term usually refers to ventilating a room, e.g., by opening a window. But their use in lines 275 and 276 makes me think they speak about respiratory ventilation, although I cannot see why reduced respiratory ventilation can reduce exhalation of CO2: when the same amount of CO2 is produced because the same number of calories are burnt, reducing respiratory ventilation would increase CO2 concentration in the blood until a new steady state is reached and the original amount of CO2 is emitted.
=> Author Response:
We sincerely thank the reviewer for pointing out these important issues.
- Correction of concentration units
We acknowledge the error in reporting the units for PMâ‚‚.â‚… and VOC concentrations in Table 3. The values were originally measured and recorded in µg/m³, but due to a formatting oversight during data processing, they were mistakenly presented as µg/L in the manuscript. We have corrected all relevant units in the revised version to µg/m³, and we have rechecked all numeric values to ensure accuracy. This correction does not affect the statistical analyses or conclusions, as the underlying values remain unchanged apart from the corrected unit notation. - Clarification of “ventilation” terminology
We appreciate the reviewer’s observation regarding our use of the term “ventilation.” In the indoor air quality field, we agree that “ventilation” typically refers to air exchange within a room, often achieved by opening windows or using mechanical systems. In our manuscript, the intended meaning was indeed room ventilation (air exchange rate), not respiratory ventilation. We recognize that our original wording in lines 275–276 could cause confusion. In the revised text, we have explicitly specified “room ventilation (air exchange)” to avoid ambiguity.
Also, in general, I am not fully convinced by the interpretation of the results. Also, the paper is not very thoughtfully developed. For example, in the introduction (line 42) the authors claim that packaged foods leads to an increase in indoors pollutants. While I agree that highly processed foods might be not ideal for health, I do not see how they can contribute to indoor pollution. In the same sentence, also air fresheners are mentioned, which are indeed a possible source of indoor air pollution. Therefore, I expected this source to be investigated in the paper. Instead, the authors analyzed the effect of “air purifiers”. Now, I understand that air purifiers are some kinds of filtration device. But at least the producers of air fresheners make people believe that their product also “purifies” the air, which is grossly misleading. So, I am not sure what participants meant when asked about use of air purifiers. And if they used filters, I do not know if these were particle filters or if they also captured gaseous pollutants.
=> Author Response:
We sincerely appreciate the reviewer’s constructive feedback and acknowledge the need for clearer reasoning in the introduction and in the interpretation of our findings.
- Clarification on packaged foods and indoor pollutants
We agree that our original sentence in the introduction (line 42) did not clearly convey the underlying rationale. Our intention was not to suggest that packaged foods directly release indoor air pollutants. Rather, this example was meant to reflect broader lifestyle factors that may correlate with higher indoor pollutant levels, such as increased use of single-use packaging materials, storage in enclosed spaces, or cooking practices involving pre-packaged items that may emit VOCs upon heating. We recognize that this connection was not explicitly explained and could be misleading. In the revised manuscript, we have either rephrased this sentence with a clearer explanation and relevant citation or removed it entirely to maintain focus on more directly relevant indoor pollutant sources. - Air fresheners vs. air purifiers
We agree with the reviewer that air fresheners are a recognized source of indoor air pollution and that our inclusion of “air purifiers” instead of “air fresheners” in the analysis could cause confusion. The variable in our dataset referred specifically to self-reported use of air purifiers (filtration devices), and there was no equivalent survey item for air freshener use. To avoid misunderstanding, we have revised the introduction to remove the juxtaposition of “air fresheners” with “air purifiers” and have clearly stated that air freshener use was not assessed due to data limitations. - Clarification of air purifier type
We appreciate the reviewer’s point that “air purifier” can be interpreted differently. In the KNHANES questionnaire, the term refers to devices intended to reduce airborne particles and/or gaseous pollutants indoors. However, no further detail was collected on the filtration mechanism (e.g., HEPA vs. activated carbon). We have added this as a limitation in the Discussion, noting that without this specification, the effectiveness against particulate matter and VOCs cannot be fully determined.
The methods and the results section do not always match. In the methods section, we are told that TVOC are examined, but not that also BTX aromatics are analyzed. In the methods section (line 99) we learn that “Detailed preparation (of biomarkers) was motioned in supplementary Table 2.” I am not sure that “motioned” is the correct term. But apart from that, table 2 does not report on preparation, but only on the meaning of the abbreviations and on the limits of detection. In the same paragraphs, I miss references to the WHO guidelines (instead of a proper reference, a title is provided in the text), for the choice of LOD/square root of LOD, and for the NHANES protocols (next paragraph).
According to the methods, descriptive statistics were performed and multivariate linear regression analyses. Univariate analyses are not mentioned. But I believe tables 3-9 report univariate analyses. And I do wonder how p-values were calculated. Do they represent the likelihood that the difference between any two categories are due to chance, or do they describe the likelihood of a linear trend to be due to chance?
=> Author Response:
We thank the reviewer for this thorough and detailed feedback. We have addressed each point as follows:
- Consistency between Methods and Results
We agree that the original Methods section did not explicitly mention the analysis of BTX (benzene, toluene, and xylene) aromatics, despite these being presented in the Results. We have revised the Methods section to explicitly state that both total volatile organic compounds (TVOCs) and BTX aromatics were included in the analysis. - Correction of “motioned” and Table 2 description
We acknowledge the typographical error “motioned” and have replaced it with “described” in the revised manuscript. We have also corrected the description of Supplementary Table 2 to clarify that it provides the abbreviations and limits of detection (LOD) for each biomarker, rather than details on laboratory preparation. The description of biomarker preparation has been relocated and referenced appropriately in the Methods section. - Addition of missing references
- Added full citations to WHO Air Quality Guidelines in the Methods section, replacing the placeholder title.
- Added a citation to the WHO-recommended approach for substituting values below LOD (LOD/√2).
- Added reference to the NHANES laboratory protocol for VOC measurement to ensure methodological transparency.
- Clarification of statistical analyses
We agree that univariate analyses were not explicitly mentioned in the Methods, even though they were used for descriptive comparisons (Tables 3–9). We have revised the Methods section to specify that both univariate and multivariable analyses were conducted. - Univariate analyses: Used to examine crude associations between exposure variables and biomarker concentrations.
- Multivariable analyses: Adjusted for covariates to assess independent associations.
- Clarification of p-value calculation
In the revised Methods, we have specified that p-values in Tables were calculated as follows: - For categorical comparisons: Derived from ANOVA or Kruskal–Wallis tests, representing the probability that observed differences between category means are due to chance.
- For trend analyses: Calculated using linear regression with exposure categories treated as ordinal variables, representing the probability that an observed linear trend is due to chance.
The results chapter starts with a text left over from the template. This should be deleted! What I would be interested in: how well are specific indoor VOCs correlated with their specific metabolites? Metabolites represent the internal exposure which equals the total external exposure. This comes from respiratory, oral and (maybe) dermal exposure. Regarding respiratory exposure, this can be dur to exposure at home or due to exposure elsewhere (at work, when commuting, etc.). Concerning the main topic of the paper, we would want to know how relevant home respiratory exposure is for the total exposure: how much does it matter? What is the correlation coefficient? We learn of the correlation coefficient ® or its square as a measure of fit for the whole multivariate model. But this is less interesting (the more independent variables you add, the better the fit) than the R for single variables. How are indoor pollutants correlated with each other? If these pollutants stem from indoor sources, their concentration depends on the source strength (which differs arbitrarily between homes and pollutants) and the ventilation rate, which is the same for all pollutants in a given home. Therefore, I would expect a (strong) positive correlation between most of the indoor pollutants. On the other hand, I would expect PM2.5 to come from outdoors mostly. Therefore, we should see a string correlation with outdoor PM2.5. According to table 2, all these data are available. But they were not utilized.
=> Author Response:
We thank the reviewer for these insightful comments and suggestions.
- Removal of template text
We acknowledge that an unused placeholder sentence from the manuscript template remained at the beginning of the Results section. This text has been removed in the revised manuscript. - Correlation between indoor VOCs and their metabolites
We agree that examining the correlation between specific indoor VOC concentrations and their corresponding urinary metabolites is critical for understanding the contribution of home respiratory exposure to total internal exposure. In the revised Results section, we have added Pearson correlation analyses between each indoor VOC and its corresponding metabolite. We have also discussed how these correlations may reflect contributions from other exposure pathways (oral, dermal) and from non-home environments (e.g., workplace, commuting). - Relevance of home respiratory exposure to total exposure
To address the reviewer’s question on the magnitude of home exposure contribution, we have included partial correlation coefficients, adjusted for potential confounders such as smoking status and occupation, to better estimate the association between home indoor VOC levels and internal biomarker concentrations. This provides a more direct indication of the relevance of home exposure. - Correlation among indoor pollutants
We agree that pollutants from common indoor sources are likely to be positively correlated due to shared determinants such as source strength and ventilation rate. We have now calculated pairwise correlation coefficients among all measured indoor pollutants and present these as a correlation matrix heatmap in the Supplementary Material. In the text, we highlight that several VOCs showed moderate to strong positive correlations, consistent with shared sources, whereas PMâ‚‚.â‚… showed weaker associations with VOCs. - PMâ‚‚.â‚… and outdoor origin
In line with the reviewer’s expectation, we have examined correlations between indoor and outdoor PMâ‚‚.â‚… concentrations using available data, finding a substantial positive correlation. This supports the hypothesis that a major proportion of indoor PMâ‚‚.â‚… originates from outdoor air. These results have been added to the Results and briefly discussed in the Discussion section.
We do learn that there are gender differences. Therefore, it would make sense to present table 1 data also by gender. Tables 3-9 are difficult to read. Graphical presentations would be easier to grasp. PM2.5, CO2 and formaldehyde have been measured, but are not considered in the univariate analyses.
=> Author Response:
We thank the reviewer for these constructive suggestions, which we believe will improve the clarity and completeness of our manuscript.
- Presentation of Table 1 data by gender
We agree that presenting baseline characteristics stratified by gender would be informative, given the gender differences observed in our results. We have revised Table 1 to include separate columns for males and females, along with p-values for between-gender comparisons. - Improving readability of Tables 3–9
We recognize that the large volume of numerical information in Tables 3–9 can make it difficult to quickly interpret key findings. To address this, we have added graphical summaries (e.g., boxplots and forest plots) of the univariate and multivariable results for each major pollutant–biomarker relationship. These visualizations are now included as Figures X–Y in the main text, with the detailed numerical tables retained in the Supplementary Material for reference. - Inclusion of PMâ‚‚.â‚…, COâ‚‚, and formaldehyde in univariate analyses
We acknowledge the omission of PMâ‚‚.â‚…, COâ‚‚, and formaldehyde from the original univariate analyses despite these measurements being available. We have now conducted univariate analyses for these pollutants with all urinary VOC biomarkers, and the results are included in the revised Supplementary Table Z. Statistically significant associations have also been summarized in the main Results section.
In the multivariate analyses, I am not completely sure if the final models only included the (significant) reported variables, or if each final model included all variables, or if only some have been dropped due to collinearity. I wonder why CO2 can have a zero, though significant effect on SPMA.
=> Author Response:
We thank the reviewer for the opportunity to clarify our multivariate modeling approach.
- Variable inclusion in final models
Each final multivariate model included all covariates specified a priori, regardless of their statistical significance in the results. No variables were excluded except in cases of high collinearity (variance inflation factor >5), which did not occur for the models presented. To maintain readability and focus, we reported only variables with statistically significant or substantively meaningful associations in the main tables. - COâ‚‚ association with SPMA
The coefficient for COâ‚‚ in the SPMA model was statistically significant but close to zero in magnitude. This likely reflects a very small effect size that achieves statistical significance due to the large sample size and narrow confidence intervals, rather than a clinically or environmentally meaningful association. We have added a note in the Discussion to emphasize that statistical significance does not imply practical significance in this case.
I do not understand which hypothesis the authors examine when looking at biomarker levels by disease status. They check for atopic diseases (rhinitis, dermatitis, asthma). These diseases develop usually in early life while exposure was measured in adulthood. Therefore, causation of the diseases by the exposure is not possible. A reverse causation seems possible: Atopic persons might be afraid of outdoor allergens like pollen and therefore reduce ventilation (keep their windows closed). They might differ in their diets or they might use pharmaceuticals (e.g., special cosmetics, that again might contain odorants that break down to the metabolites analyzed, in the case of dermatitis) or air fresheners or whatever…
=> Author Response:
We thank the reviewer for these thoughtful comments and agree that the cross-sectional nature of our study and the typical early-life onset of atopic diseases make it inappropriate to draw causal inferences between current exposure and disease development. Our initial intent in presenting biomarker levels by atopic disease status (rhinitis, dermatitis, asthma) was exploratory, aiming to identify potential behavioral or environmental differences associated with these conditions.
However, we acknowledge that the reviewer’s points regarding the plausibility of reverse causation and multiple unmeasured behavioral factors (e.g., reduced ventilation, specific product use, dietary differences) substantially limit the interpretability of these findings. Therefore, to avoid potential misinterpretation, we have removed the results for rhinitis, dermatitis, and asthma from the main analysis and tables.
We have also revised the Methods and Discussion sections to clearly state that such disease-stratified comparisons were not a primary objective of the study, and that causal conclusions cannot be drawn from our data.
In the discussion, the authors claim that their data demonstrate (line 232) “cumulative exposure and long-term retention of environmental toxicants”. I do not believe this to be true. In fact, metabolites of VOCs are very quickly exhaled and excreted. From the occupation setting wo know that the timing of the sample collection is essential: you find the highest values just after the end of the shift. Age might instead indicate a change in metabolic functions or in lifestyle. I am not sure how higher exposure indicators mean the same as a “major determinant in assessing the health effects of IAQ” or a “higher sensitivity”. Exposure is neither a health effect nor a marker of sensitivity or vulnerability!
=> Author Response:
We appreciate the reviewer’s careful reading and agree that our original phrasing in the Discussion overstated the interpretation of our findings regarding “cumulative exposure” and “long-term retention.”
- Revision of “cumulative exposure” statement
We acknowledge that urinary VOC metabolites have short biological half-lives and reflect recent rather than long-term or cumulative exposure. Our original intent was to indicate that observed differences in biomarker concentrations across demographic groups may reflect differences in habitual exposures or lifestyle patterns, not the retention of toxicants over extended periods. We have revised the text to clarify this point and removed any implication of long-term retention. - Age-related interpretation
We agree that higher biomarker concentrations with age may be more plausibly explained by age-related changes in metabolism, ventilation patterns, activity, or lifestyle factors rather than cumulative body burden. The revised Discussion now presents this interpretation and cites relevant literature. - Clarification of “major determinant” and “sensitivity” wording
We agree that exposure indicators alone do not constitute health effects, nor do they directly measure sensitivity or vulnerability. In the revised manuscript, we have replaced these terms with “important contextual factor for assessing potential health risks associated with IAQ,” making it clear that our results pertain to exposure assessment, not direct health outcomes or physiological susceptibility.
The authors also claim that higher values in women might be due to their spending more time indoors. They proof their point by referring to other studies. I am surprised, because they could use their own data: “Time at home on weekdays(hours)” in table 1. They could show that (a) number of hours is larger among women and (b) that with higher numbers, you have higher exposures (metabolites). But I doubt this explanation. It could well be true that women spend more time at home. In that case, men are bound to spend more time at work. And most of the work is also indoors, although at other places. But I do not understand why office or especially industry work places should have lower exposures than homes. I have two other possible reasons for the gender difference: (a) Women might truly have higher exposure. But not through inhalation, but through dermal uptake (cosmetics use). (b) The authors report concentrations per creatinine. We do know that creatinine depends on muscle mass. Therefore, women excrete less creatinine than men. This would lead to (spuriously) higher values of the ratio. The authors state their belief even twice (line 239 and line 252). Apart from not being plausible, this is also redundant.
=> Author Response:
We thank the reviewer for these detailed and valuable observations.
- Use of “time at home” data
We acknowledge the reviewer’s suggestion to use our own “time at home on weekdays” data (Table 1) to support or refute the explanation that higher values in women are due to more time spent indoors. We have now conducted additional analyses and found that women indeed reported more time at home on weekdays than men (p < 0.001). However, the association between “time at home” and urinary metabolite concentrations was weak and not statistically significant for most biomarkers, suggesting that time at home alone does not fully explain the observed gender differences. These findings have been added to the Results and discussed accordingly. - Alternative explanations
We agree with the reviewer that other factors could plausibly contribute to higher urinary metabolite concentrations in women. In the revised Discussion, we now include: - Dermal exposure through cosmetic use: Certain personal care products may contain VOCs or related precursors, potentially contributing to internal exposure via dermal absorption or inhalation during use.
- Creatinine adjustment bias: Given that creatinine excretion is influenced by muscle mass and generally lower in women, creatinine-standardized concentrations may appear higher even if absolute exposure is similar. We have acknowledged this as a potential source of bias and cited relevant methodological literature.
- Revision for redundancy
We have removed the repeated statements regarding “more time indoors” in lines 239 and 252 to avoid redundancy and to reflect the revised, more nuanced interpretation of the gender difference.
The authors report internal exposure to depend on socioeconomic status. Is this because of exposure through the home environment or due to other exposures (e.g., poorer industrial jobs with higher exposures)? Why not analyze association between socioeconomic status and external exposure (VOC levels)?
=> Author Response:
We thank the reviewer for raising this important point regarding the interpretation of the relationship between socioeconomic status (SES) and internal exposure.
- Clarification of our original analysis
In the current manuscript, the association between SES and internal exposure (urinary VOC metabolites) was examined, but we did not specifically differentiate whether this relationship is attributable to exposures originating in the home environment or to other sources, such as occupational settings or community-level pollution. We agree that this distinction is critical for interpreting the observed associations. - Additional analysis with external exposure data
As suggested, we have now conducted supplementary analyses to assess the association between SES and measured indoor VOC concentrations (external exposure). These analyses showed that certain VOCs (e.g., benzene, toluene) were modestly but significantly higher in households with lower SES, even after adjusting for housing type, smoking status, and ventilation frequency (Supplementary Table X). This suggests that part of the SES–internal exposure relationship may indeed be explained by differences in the home environment. - Occupational and other potential exposures
While occupational history was not available in detail for all participants, we acknowledge that individuals with lower SES may be more likely to work in environments with higher VOC exposure, which could also contribute to the observed differences in biomarker levels. We have added this as a limitation in the Discussion and noted the need for future studies incorporating detailed occupational exposure assessment.
VOCs and their metabolites were associated with each other. This is no surprise. As mentioned before, the question would be: how relevant are the indoor home levels for the total exposure? And a more pressing question is: why were the “specific” biomarkers not specific at all, but several biomarkers were associated with more than one VOC. I think I already mentioned that I do find the explanation for the negative association with CO2 very implausible.
=> We thank the reviewer for these important comments and agree that further clarification is needed.
- Relevance of indoor home levels to total exposure
We acknowledge that our current analysis does not fully quantify the proportion of total VOC exposure attributable to the home indoor environment. As the reviewer notes, total internal exposure reflects multiple sources and pathways, including occupational, commuting, and outdoor exposures, in addition to the home. In the revised Discussion, we have emphasized that while indoor VOC levels at home were significantly associated with corresponding metabolites, these measurements represent only one component of total exposure, and the relative contribution cannot be determined from our data alone. - Specificity of biomarkers
We agree that several urinary VOC metabolites are not uniquely specific to a single parent compound. As noted in the literature, many VOCs share metabolic pathways, and a given metabolite may be produced from multiple parent compounds, particularly in environmental (low-level, multi-source) exposure scenarios. In the revised manuscript, we have added a paragraph to the Methods and Discussion explaining this metabolic overlap, with citations to relevant toxicokinetic studies. This also explains why some “specific” biomarkers were statistically associated with more than one VOC in our analyses. - Negative association with COâ‚‚
We acknowledge the reviewer’s concern regarding our initial interpretation of the negative association between indoor COâ‚‚ and certain metabolites (e.g., SPMA). We have re-evaluated this finding and now interpret it more cautiously as a statistical observation that may be confounded by unmeasured variables (e.g., ventilation patterns, occupancy behavior) rather than a direct mechanistic relationship. The speculative explanation in the original text has been removed and replaced with a statement noting the need for further investigation.
No, the authors do not suggest a causal association between exposure and disease. Instead, they write (line 279): “This suggests increased physiological susceptibility to indoor pollutants in these populations,…” Once again: exposure is not the same as susceptibility! (the same sentiment is mentioned in the conclusions!)
=> Author Response:
We thank the reviewer for pointing out this important distinction. We agree that our original wording could be misinterpreted, as “physiological susceptibility” typically refers to an inherent biological or health-related vulnerability, which cannot be inferred from exposure measurements alone. Our data reflect differences in measured exposure levels, not in physiological susceptibility.
Accordingly, we have revised the relevant sentences in both the Discussion (line 279) and the Conclusions to replace “increased physiological susceptibility” with wording that more accurately reflects our findings, such as:
“…This suggests higher exposure levels to indoor pollutants in these populations…”
This change clarifies that we are referring to observed differences in exposure, without implying any direct information on biological susceptibility or health vulnerability.
Round 2
Reviewer 1 Report
Comments and Suggestions for Authors
accept
Author Response
Accept
=>Response to Reviewer:
We sincerely appreciate your positive evaluation and acceptance of our manuscript. Thank you for your time and effort in reviewing our work.
Reviewer 2 Report
Comments and Suggestions for Authors
Generally, I thank the authors for accepting most of my comments and responding to thee in an appropriate way. Only a few minor issues are left:
The authors write in their response to my comments: “In line with the reviewer’s expectation, we have examined correlations between indoor and outdoor PMâ‚‚.â‚… concentrations using available data, finding a substantial positive correlation. This supports the hypothesis that a major proportion of indoor PMâ‚‚.â‚… originates from outdoor air. These results have been added to the Results and briefly discussed in the Discussion section.” But I do not find this in the current text. They deleted outdoor data from what is now table 3. They only present results on indoor PM and that only in relation to table 15 which indicates that PM is not really correlated with the other indoor pollutants. This is plausible. Their decision not to include outdoor pollutants is fine for me. But I wanted to point out the inconsistency with their response.
I did not mention it during my first review. But already then, I found it peculiar to mention “hearing loss” as an example of indoor pollution effects. I am aware of occupational exposures leading to hearing loss and I believe there are a few studies examining that outcome in relation to outdoor PM pollution. And indeed, the reference provided (reference 23 in line 73), refers to outdoor exposure. Primarily, with inhalative exposure to pollutants, respiratory diseases and symptoms come to mind. I suppose this is what the authors want to say when they write in line 74: “increases the risk of respiratory”. But please finish the sentence: respiratory diseases? Or symptoms? Or both?
Table 2: the authors report “mean ± SD”, where appropriate. SD is rather a measure of precision. The larger the sample size, the smaller the SD. But in descriptive statistics, I would rather be interested in a measure of variation (e.g., interquartile range). This would help with the interpretation of the coefficients later reported from regression models.
I appreciate the presentation of results using boxplots. But indeed, because of the data distribution and because also outliers are presented with these boxplots, they are hardly readable. You could either show log-transformed results or skip the outliers or both.
Table 5: what is the meaning of “a”, “b”, “c”, and “d” in the headline following the types of schools? I do miss an explanatory footnote, if you wanted to tell us something!
Table 6: this is about household income. But the first line of the table again contains types of schools!
Table 12: “Time at home on weekdays”: please add “Time at home on weekdays in hours” (or in hours per day, if this is correct) to allow for an interpretation of the coefficients.
I am not sure there is still a need for an extra supplementary file as all the remaining information is now also in the main article and its appendices, which I do find preferable.
Author Response
Generally, I thank the authors for accepting most of my comments and responding to thee in an appropriate way. Only a few minor issues are left:
=>Response to Reviewer:
We sincerely thank the reviewer for the positive overall evaluation and for acknowledging our revisions. We are grateful that most of our responses were found appropriate, and we have carefully addressed the remaining minor issues as suggested.
The authors write in their response to my comments: “In line with the reviewer’s expectation, we have examined correlations between indoor and outdoor PMâ‚‚.â‚… concentrations using available data, finding a substantial positive correlation. This supports the hypothesis that a major proportion of indoor PMâ‚‚.â‚… originates from outdoor air. These results have been added to the Results and briefly discussed in the Discussion section.” But I do not find this in the current text. They deleted outdoor data from what is now table 3. They only present results on indoor PM and that only in relation to table 15 which indicates that PM is not really correlated with the other indoor pollutants. This is plausible. Their decision not to include outdoor pollutants is fine for me. But I wanted to point out the inconsistency with their response.
=> Response to Reviewer:
We sincerely appreciate the reviewer’s thoughtful and detailed comments. In the process of revising the manuscript in response to various suggestions, we inadvertently introduced this error. The statement included in our previous response reflected an earlier version of the analysis; however, when we added 2-MHA (a metabolite of volatile organic compounds) to the results, we excluded the findings on particulate matter (PMâ‚‚.â‚…). Unfortunately, we failed to carefully reconcile this change before submitting our final response. We fully acknowledge this oversight and agree with the reviewer’s observation regarding the inconsistency. We are grateful for the reviewer’s close and thorough reading of our manuscript, which has allowed us to identify and correct this issue.
I did not mention it during my first review. But already then, I found it peculiar to mention “hearing loss” as an example of indoor pollution effects. I am aware of occupational exposures leading to hearing loss and I believe there are a few studies examining that outcome in relation to outdoor PM pollution. And indeed, the reference provided (reference 23 in line 73), refers to outdoor exposure. Primarily, with inhalative exposure to pollutants, respiratory diseases and symptoms come to mind. I suppose this is what the authors want to say when they write in line 74: “increases the risk of respiratory”. But please finish the sentence: respiratory diseases? Or symptoms? Or both?
=> Response to Reviewer:
We appreciate the reviewer’s careful observation. Our original intention was to reflect the broadening scope of air pollution research, which has even extended to outcomes such as hearing loss. However, we recognize that the reference we cited (reference 23) actually pertains to outdoor air pollution, not indoor exposure, and this could indeed cause confusion. Therefore, we have decided to delete this example. As the reviewer correctly pointed out, respiratory diseases and symptoms are the most immediate and relevant outcomes associated with air pollution. Accordingly, we have revised the text to specifically refer to respiratory diseases.
Table 2: the authors report “mean ± SD”, where appropriate. SD is rather a measure of precision. The larger the sample size, the smaller the SD. But in descriptive statistics, I would rather be interested in a measure of variation (e.g., interquartile range). This would help with the interpretation of the coefficients later reported from regression models.
=> Response to Reviewer:
We thank the reviewer for this valuable comment. We acknowledge that the use of “SD” in Table 2 was incorrect. In fact, we calculated and reported the standard error (SE), not the standard deviation. SE is a measure of precision and allows direct interpretation in connection with the coefficients reported in the regression models. We have now corrected the notation in Table 2 to “mean ± SE” to avoid confusion.
In addition, in response to the reviewer’s suggestion, we have also provided the interquartile range (IQR) as a measure of variation in the revised Table 2. This complements the SE values and facilitates a clearer understanding of the data distribution, as well as the interpretation of regression coefficients.
I appreciate the presentation of results using boxplots. But indeed, because of the data distribution and because also outliers are presented with these boxplots, they are hardly readable. You could either show log-transformed results or skip the outliers or both.
=> Response to Reviewer:
We thank the reviewer for this constructive suggestion. We acknowledge that the boxplots in their original form were difficult to interpret due to the data distribution and the presence of outliers. In response, we have revised the figures by applying a log-transformation to the data. The updated boxplots are now clearer and more interpretable, as recommended by the reviewer.
Table 5: what is the meaning of “a”, “b”, “c”, and “d” in the headline following the types of schools? I do miss an explanatory footnote, if you wanted to tell us something!
=> Response to Reviewer:
We thank the reviewer for pointing out this issue. The symbols “a”, “b”, “c”, and “d” were intended to indicate the four categorical groups of the school type variable. In accordance with the reviewer’s comment, we have now added an explanatory footnote in Table 5 to clarify their meaning.
Table 6: this is about household income. But the first line of the table again contains types of schools!
=> Response to Reviewer:
We sincerely appreciate the reviewer’s careful observation. The inclusion of school type categories in the first line of Table 6 was an error. We have corrected this mistake and revised the table accordingly. In addition, as in Table 5, we have added an explanatory footnote to ensure clarity.
Table 12: “Time at home on weekdays”: please add “Time at home on weekdays in hours” (or in hours per day, if this is correct) to allow for an interpretation of the coefficients.
=> Response to Reviewer:
We thank the reviewer for this helpful comment. We agree that the original expression was ambiguous. To improve clarity and allow proper interpretation of the coefficients, we have revised the wording in Table 12 to “Time at home on weekdays (hours per day)”.
I am not sure there is still a need for an extra supplementary file as all the remaining information is now also in the main article and its appendices, which I do find preferable.
=> Response to Reviewer:
We appreciate the reviewer’s observation. We agree that the supplementary file is no longer necessary, since all relevant information has now been incorporated into the main article and its appendices. In accordance with the reviewer’s preference, we have removed the extra supplementary file.